# iFusion: Integrating Dynamic Interest Streams via Diffusion Model for Click-Through Rate Prediction

**Ziheng Ni**[1][*]**, Congcong Liu**[1][*][†]**, Yuying Chen**[2]**, Zhiwei Fang**[1]**, Changping Peng**[1]**,**
**Zhangang Lin**[1]**, Ching Law**[1]**, Jingping Shao**[1]
[1] JD.com    [2] The Hong Kong University of Science and Technology
niziheng@stu.pku.edu.cn, cliubh@connect.ust.hk

## Abstract

Click-through rate (CTR) prediction is crucial for recommendation systems and online advertising, relying heavily on effective user behavior modeling. While existing methods separately refine long-term and short-term interest representations, the fusion of these behaviors remains a critical yet understudied challenge due to misaligned feature spaces, disjointed modeling, and noise propagation in short-term interests. To address these limitations, we propose iFusion, a diffusion-based generative user interest fusion method, which reformulates interest fusion as a conditional generation process. iFusion leverages short-term interests as conditional guidance and progressively integrates long-term representations through denoising, eliminating reliance on linear fusion assumptions. Our framework introduces two key components: (1) the Disentangled Classifier-Free Diffusion Guidance (DCFG) Mechanism, which adaptively disentangles core preferences from transient fluctuations, and (2) the Mixture AutoRegressive Denoising Network (MARN), which enables joint interest modeling and fusion through autoregressive denoising. Experiments demonstrate that iFusion outperforms baselines across public and industrial datasets, as well as in online A/B tests, validating its effectiveness in robust CTR prediction. This work establishes a new paradigm for generative user interests fusion in CTR prediction.

## 1 Introduction

Dynamic interest fusion stands as a fundamental challenge in click-through rate (CTR) predictionZhou et al. (2019); Chen et al. (2019); Zhou et al. (2018); Liu et al. (2023a); Ni et al. (2025); Sang et al. (2025a;b); Zhu et al. (2023; 2024) in recommendation systems and online advertising. At the core of this challenge lies the effective fusion of heterogeneous behavioral signals across different temporal scales particularly the integration of stable long-term preferences with volatile short-term interests.

The prevailing approach partitions user behaviors into long-term and short-term sequences (Chen et al., 2019), modeling them separately before fusion. Despite advances in individual components (Pi et al., 2020; Chang et al., 2023; Si et al., 2024; Hidasi et al., 2016; Xia et al., 2023), interest fusion remains critically under-explored, particularly for handling non-stationary and contradictory evolution patterns. Current fusion methods including concatenation (Zhou et al., 2018), attention mechanisms (Vaswani et al., 2017; Li et al., 2019), and gating networks (Hochreiter & Schmidhuber, 1997; Lv et al., 2019) rely on linear assumptions that prove inadequate, suffering from three key limitations: First, misaligned heterogeneous feature spaces arise from divergent feature selection mechanisms. (Shen et al., 2022; Zheng et al., 2022) Deployment constraints often enforce distinct representations for long- and short-term interest modeling. For instance, long-term behaviors are modeled using historical click logs (Pi et al., 2020; Si et al., 2024), whereas short-term behaviors leverage purchase interactions (Zhou et al., 2018). This inherent heterogeneity creates non-linear interactions between temporal scales that conventional linear fusion operators fail to capture, as they

---

[*]Equal contribution.
[†]Corresponding authors.

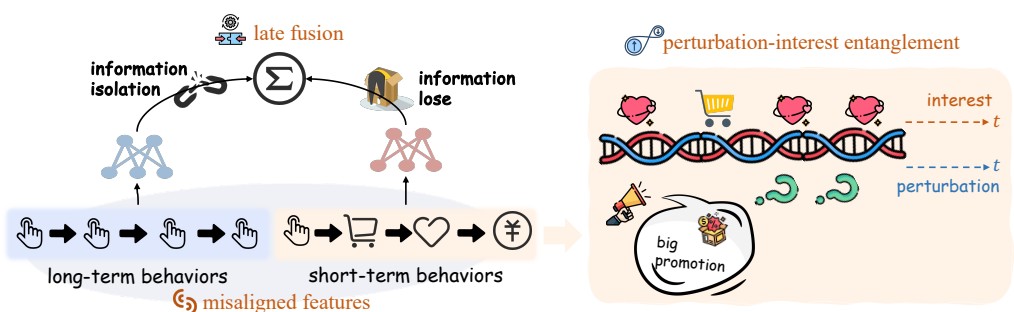

Figure 1: Inherent limitations of current interest fusion methods

presuppose feature space alignment. Second, the limitations of disjointed modeling in late-fusion paradigms. Existing late-fusion methods (He et al., 2023) employ decoupled pipelines that isolate behavior modeling from interest fusion, resulting in suboptimal inductive biases that hinder effective cross-sequence integration. Third, perturbation-interest entanglement. Linear fusion lacks explicit mechanisms to disentangle meaningful interest signals from stochastic fluctuations in short-term behaviors, allowing noise to propagate unchecked and corrupt stable long-term representations. Moreover, HSTU-based generative solutions (Zhai et al., 2024; Han et al., 2025) for ranking tasks intensify reliance on the richness of sequential information by constructing user historical behaviors into one unified fusion sequence. When behavioral data is sparse, this approach leads to inadequate modeling performance due to its inherently discriminative nature in ranking tasks, which fails to effectively infer and fuse interests from limited behavioral evidence.

To address these challenges, we introduce iFusion, a novel framework that reformulates interest fusion as a conditional generation process rather than a deterministic fusion operation. Our approach leverages diffusion models to gradually integrate long-term interest representations conditioned on short-term behavioral guidance through a structured denoising process. This generative formulation eliminates restrictive linear assumptions, enabling more flexible and robust interest modeling. The iFusion framework incorporates two synergistic components: the Disentangled Classifier-Free Guidance (DCFG) Mechanism, which explicitly separates core preference signals from transient fluctuations during the guidance process, and the Mixture AutoRegressive Denoising Network (MARN), which enables independent conditioning of multiple denoising pathways while capturing fine-grained interest evolution. Together, these components provide a unified solution to the fundamental challenges of interest fusion. Our main contributions are summarized as follows:

- **Generative Reformulation of Interest Fusion:** We propose a novel diffusion-based framework for CTR prediction that treats interest fusion as a conditional generation process, moving beyond traditional deterministic fusion paradigms.

- **Joint Interest Modeling via MARN:** Leveraging the proposed MARN, we simultaneously model the fusion of long- and short-term interests while capturing fine-grained interest evolution within short-term behavioral sequences.

- **Disentangled Guidance with DCFG:** By adopting the DCFG paradigm, we disentangle supervision signals to explicitly quantify the contributions of core preference modeling and transient behavioral fluctuations.

- **Comprehensive Empirical Validation:** We demonstrate the effectiveness of iFusion through extensive experiments on public benchmarks, industrial datasets, and online A/B tests with CTR prediction serving as a key evaluation scenario.

## 2 RELATED WORK

### 2.1 DISCRIMINATIVE USER BEHAVIOR MODELING

User behavior modeling plays a pivotal role in capturing user preferences, with widespread applications in recommendation systems and online advertising (He et al., 2023). Early approaches primarily

relied on MLP architectures (Wang et al., 2015), later evolving to sequence-aware models such as RNNs (Hidasi et al., 2015; Hidasi & Karatzoglou, 2018; Li et al., 2017; Quadrana et al., 2017) and attention mechanisms (Vaswani et al., 2017; Zhou et al., 2018; 2019). Recent advances focus on modeling long-term dependencies (Ren et al., 2019; Pi et al., 2019; 2020; Chen et al., 2021; Cao et al., 2022), handling heterogeneous behavior types (Xia et al., 2021; 2020; Meng et al., 2020), and integrating contextual side information (Li et al., 2020; Lei et al., 2021; Liu et al., 2021; Xie et al., 2022).

## 2.2 GENERATIVE MODELING METHODS

Generative models have prompted a paradigm shift in recommendation systems, transitioning from discriminative to generative approaches (Wei et al., 2025; Guo et al., 2025; Zhou et al., 2025a;b; Zhai et al., 2024; Han et al., 2025). In particular, diffusion models have gained significant traction in this domain (Yang et al., 2023; Li et al., 2023; Du et al., 2023; Niu et al., 2024; Wang et al., 2024; 2023). However, existing diffusion-based work primarily focuses on sequential recommendation tasks, with limited exploration of generative paradigms for click-through rate prediction (Lai et al., 2025), which requires effective modeling of dynamic user interest fusion, representing a key limitation in the current literature.

## 3 PRELIMINARY

For each user, we consider two complementary behavioral sequences that capture different temporal granularities of user interests. The *long-term behavior sequence* $\mathcal{B}^{\mathrm{L}} = \{b_1^{\mathrm{L}}, b_2^{\mathrm{L}}, \ldots, b_M^{\mathrm{L}}\}$ represents the user's historical interaction patterns over an extended period, where $M$ denotes the sequence length. The *short-term behavior sequence* is organized into sessions $\mathcal{B}^{\mathrm{S}} = \{S_1, S_2, \ldots, S_K\}$, where each session $S_i = \{b_1^{\mathrm{S}}, b_2^{\mathrm{S}}, \ldots, b_{n_i}^{\mathrm{S}}\}$ contains interactions within a confined temporal window, and $K$ is the total number of sessions. We employ two specialized encoders to extract meaningful representations from these sequences. A long-term behavior encoder $f_{\mathrm{L}}(\cdot)$ that maps $\mathcal{B}^{\mathrm{L}}$ to a dense representation $\mathbf{h}^{\mathrm{L}} \in \mathbb{R}^{d^{\mathrm{L}}}$. And a short-term session encoder $f_{\mathrm{S}}(\cdot)$ that processes each session $S_i$ to generate session-level embeddings $\mathbf{h}_i^{\mathrm{S}} \in \mathbb{R}^{d^{\mathrm{S}}}$. Our objective is to learn a fusion function $\mathcal{F}_\theta$ that generates an enriched behavior representation:

$$\mathbf{h}^{\mathrm{fusion}} = \mathcal{F}_\theta(\mathbf{h}^{\mathrm{L}}, \{\mathbf{h}_i^{\mathrm{S}}\}_{i=1}^K) \tag{1}$$

where $\theta$ denotes learnable parameters. The fused representation $\mathbf{h}^{\mathrm{fusion}}$ should simultaneously preserve long-term user preferences and capture short-term dynamic interests for improved CTR prediction.

We combine this objective with diffusion model and reformulate the interest fusion process into a Markov chain with $T$ diffusion steps. In the **forward process**, the long-term interest $\mathbf{h}^{\mathrm{L}}$ serves as the initial state $x_0$, and gradually adds Gaussian noise according to a variance schedule $\{\beta_t\}_{t=1}^T$:

$$q(\mathbf{x}_t|\mathbf{x}_{t-1}) = \mathcal{N}(\mathbf{x}_t; \sqrt{1 - \beta_t}\mathbf{x}_{t-1}, \beta_t\mathbf{I}), \quad q(\mathbf{x}_{1:T}|\mathbf{x}_0) = \prod_{t=1}^T q(\mathbf{x}_t|\mathbf{x}_{t-1}) \tag{2}$$

We can directly sample $\mathbf{x}_t$ at any time step $t$ in closed form according to the notable property mentioned in (Ho et al., 2020):

$$q(\mathbf{x}_t|\mathbf{x}_0) = \mathcal{N}(\mathbf{x}_t; \sqrt{\bar{\alpha}_t}\mathbf{x}_0, (1 - \bar{\alpha}_t)\mathbf{I}) \tag{3}$$

where $\alpha_t := 1 - \beta_t$ and $\bar{\alpha}_t := \prod_{s=1}^t \alpha_s$. The signal-to-noise ratio $\mathrm{SNR}(t) = \bar{\alpha}_t/(1 - \bar{\alpha}_t)$ decreases monotonically with $t$, ensuring $\mathbf{x}_T$ converges to standard Gaussian noise. The **reverse process** operates as a denoising procedure that progressively refines the noisy long-term interest representation $\hat{\mathbf{x}}_T$ across $T$ timesteps. Notably, we condition the denoising steps on the short-term interest signals $\{\mathbf{h}_i^{\mathrm{S}}\}_{i=1}^K$. This ultimately generates an interest representation $\hat{\mathbf{x}}_0$ that integrates long-term and short-term interests. The reverse Markov chain is implemented by a neural network $f_\theta$, formulated as:

$$p_\theta(\hat{\mathbf{x}}_{t-1}|\hat{\mathbf{x}}_t, \{\mathbf{h}_i^{\mathrm{S}}\}_{i=1}^K) = \mathcal{N}\left(\hat{\mathbf{x}}_{t-1}; \mu_\theta\left(\hat{\mathbf{x}}_t, t, \{\mathbf{h}_i^{\mathrm{S}}\}_{i=1}^K\right), \Sigma_\theta(\hat{\mathbf{x}}_t, t, \{\mathbf{h}_i^{\mathrm{S}}\}_{i=1}^K)\right), \tag{4}$$

---

For simplicity, we set $\hat{\mathbf{x}}_T = \mathbf{x}_T$

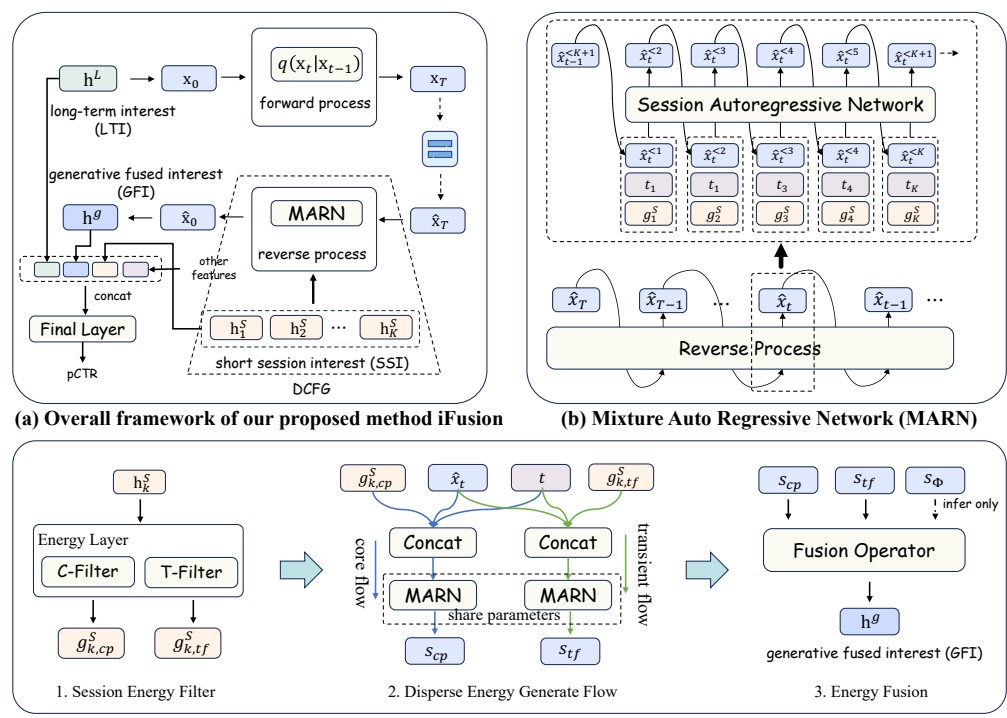

Figure 2: The framework of our proposed method iFusion

where the mean function $\mu_\theta$ integrates short-term interest guidance while the variance $\Sigma_\theta$ is typically fixed as $\sigma_t^2\mathbf{I}$ with $\sigma_t^2 = \beta_t$ or learned. Following (Ho et al., 2020), we implement the mean prediction through noise reparameterization:

$$\mu_\theta(\hat{\mathbf{x}}_t, t, \{\mathbf{h}_i^S\}_{i=1}^K) = \frac{1}{\sqrt{\alpha_t}}\left(\hat{\mathbf{x}}_t - \frac{1-\alpha_t}{\sqrt{1-\bar{\alpha}_t}}\epsilon_\theta(\hat{\mathbf{x}}_t, t, \{\mathbf{h}_i^S\}_{i=1}^K)\right), \qquad (5)$$

where $\epsilon_\theta$ serves as the guided noise predictor trained to estimate the injected noise at step $t$ while being conditioned on short-term interest signals. We utilize an alternative reparameterization approach that predicts the generated interest representation $\hat{\mathbf{x}}_0$ instead of estimating the additive noise $\epsilon$ as shown in the equation 5:

$$\mu_\theta(\hat{\mathbf{x}}_t, t, \{\mathbf{h}_i^S\}_{i=1}^K) = \sqrt{\bar{\alpha}_{t-1}}f_\theta(\hat{\mathbf{x}}_t, t, \{\mathbf{h}_i^S\}_{i=1}^K) + \frac{\sqrt{\alpha_t}(1-\bar{\alpha}_{t-1})}{\sqrt{1-\bar{\alpha}_t}}\boldsymbol{\epsilon}, \qquad (6)$$

## 4    METHODOLOGY

Our proposed iFusion method effectively addresses the key challenges outlined in Section 1. As illustrated in Figure 2, iFusion integrates multi-sequence user interests via a generative approach, collaborating with discriminative modules to enhance downstream CTR prediction. It comprises two core components: the DCFG mechanism, which provides robust guidance under perturbation for interest fusion, and the MARN, facilitating blended guidance throughout the inverse velocity field learning process.

### 4.1    DCFG: DISENTANGLED CLASSIFIER-FREE GUIDANCE FOR INTEREST FUSION

CTR prediction face a fundamental trade-off between *signal fidelity* and *perturbation robustness* when modeling user behavior sequences. While conventional classifier-free guidance (CFG) (Ho & Salimans, 2022) enables conditional generation, its uniform scaling approach proves suboptimal

for behavior modeling where signals exhibit multi-scale characteristics from stable preferences to transient fluctuations.

**Revisiting Classifier-Free Guidance.** Given short-term session interest $\mathbf{h}_i^S$ as guidance $g$, standard CFG blends conditional and unconditional predictions with a single scaling factor $\gamma$:

$$\hat{f}_\theta(\mathbf{x}_t, t, g) = f_\theta(\mathbf{x}_t, t) + \gamma(f_\theta(\mathbf{x}_t, t, g) - f_\theta(\mathbf{x}_t, t)) \tag{7}$$

This formulation assumes homogeneous signal quality across behavioral contexts, an assumption frequently violated in practice. The guidance based on interest representation has a lower signal-to-noise ratio than the guidance used in traditional image generation tasks.

**Energy-Based Perspective on Interest Dynamics.** Drawing from stochastic thermodynamics (Seifert, 2012), we model user interest dynamics as particles in a composite potential field $V(\mathbf{x}_t|g) = V(\mathbf{x}_t|g_{cp}) + V(\mathbf{x}_t|g_{tf})$, where core preferences $g_{cp}$ create deep attractors and transient fluctuations $g_{tf}$ generate shallow perturbations:

$$d\mathbf{x}_t = -[\nabla V(\mathbf{x}_t|g_{cp}) + \nabla V(\mathbf{x}_t|g_{tf})]dt + \sqrt{2D}dW_t \tag{8}$$

This analogy motivates our dual-component decomposition through functional separation. We implement disentanglement through specialized architectures capturing complementary signal aspects:

$$\mathbf{h}_{cp} = \text{AvgPool}(\text{Encoder}(g)) \quad \textit{(low-pass filtering, C-Filter)} \tag{9}$$
$$\mathbf{h}_{tf} = \text{Attention}(\text{Encoder}(g)) \quad \textit{(high-pass filtering, T-Filter)} \tag{10}$$

The core preference pathway employs strong regularization and global pooling for stability, while the transient pathway uses attention mechanisms to capture variations.

**Generalized Energy-Based Formulation.** We formulate conditional generation as sampling from an energy-based model with structurally distinct components:

$$p(\mathbf{x}_t|g) \propto \exp(-E(\mathbf{x}_t|g)) = \exp(-[\gamma_{cp}E_{cp}(\mathbf{x}_t|g) + \gamma_{tf}E_{tf}(\mathbf{x}_t|g)]) \tag{11}$$

**Theorem 1 (Energy-Based Disentanglement of Guidance)** *Let* $E(\mathbf{x}_t|g) = \gamma_{cp}E_{cp}(\mathbf{x}_t|g) + \gamma_{tf}E_{tf}(\mathbf{x}_t|g)$ *be the total energy function, where* $E_{cp}$ *and* $E_{tf}$ *are implemented through architecturally constrained networks with low-pass and high-pass characteristics respectively. Then the conditional score function admits the exact decomposition:*

$$\nabla_{\mathbf{x}_t} \log p(\mathbf{x}_t|g) = \gamma_{cp}(-\nabla_{\mathbf{x}_t}E_{cp}) + \gamma_{tf}(-\nabla_{\mathbf{x}_t}E_{tf}) \tag{12}$$

*Furthermore, if the architectural constraints enforce that the Hessians* $\nabla_{\mathbf{x}_t}^2 E_{cp}$ *and* $\nabla_{\mathbf{x}_t}^2 E_{tf}$ *have approximately orthogonal dominant eigenspaces, then the guidance directions become functionally disentangled:*

$$\langle -\nabla_{\mathbf{x}_t}E_{cp}, -\nabla_{\mathbf{x}_t}E_{tf} \rangle \leq \zeta \tag{13}$$

*where* $\zeta$ *quantifies residual correlation. (Proof in Appendix K)*

Theorem 1 provides the theoretical foundation for DCFG, demonstrating that architectural constraints induce functional disentanglement without requiring strict conditional independence. Through Fokker-Planck analysis of the Langevin system (Appendix D), we establish DCFG formulation derived based on Bayes' theorem in Appendix C:

$$\hat{f}_\theta(\mathbf{x}_t, t, g) = f_\theta(\mathbf{x}_t, t) + \sum_{j \in \{cp, tf\}} \gamma_j(f_\theta(\mathbf{x}_t, t, g_j) - f_\theta(\mathbf{x}_t, t)) \tag{14}$$

## 4.2 MARN: MIXTURE AUTO REGRESSIVE NETWORK

Current applications of diffusion models in recommendation predominantly rely on single-vector guidance during the reverse process (Yang et al., 2023), typically implemented through non-autoregressive (NAR) architectures such as MLPs or Transformers. However, NAR methods might suffer from the inability to capture fine-grained sequential dependencies due to their parallel generation nature, leading to potential inconsistency or suboptimal performance in tasks where require strict temporal

coherence (Gu et al., 2017; Kasai et al., 2020) or the guidance does not follow a completely linear relationship. Additionally, they may lack interpretability in modeling step-by-step decision processes compared to autoregressive alternatives (Stern et al., 2019). Inspired by the success of autoregressive (AR) modeling in sequential recommendation systems, we propose an autoregressive-based structure MARN for the reverse process. Unlike parallel injection methods that concatenate session embeddings or use pooling methods, MARN processes $K$ short-term session interests sequentially through chain-rule conditioning with the output for each session serving as noisy representation for the next session's generation. Compared with the NAR method, MARN has better representation learning ability, more stable training and weight adaptation advantages, which are summarized in Theorem 2, detail proof provided in Appendix E.

**Theorem 2 (AR Superiority in Multi-Session Diffusion)** *Autoregressive (AR) injection strictly dominates non-autoregressive (NAR) for multi-session diffusion with dependent sessions ($\exists i, j$ : $I(s_i; s_j) > 0$), achieving: (1) tighter KL-bound $D_{\mathrm{KL}}^{\mathrm{NAR}} - D_{\mathrm{KL}}^{\mathrm{AR}} \geq \frac{1}{2} \sum (\alpha_k - \frac{1}{K})^2 - \sum_{i<j} I(s_i; s_j)$; (2) $\mathcal{O}(K)$ lower gradient variance $\mathrm{Var}(\nabla \mathcal{L}_{\mathrm{AR}}) \leq L^2 K^{-1} \mathrm{Var}(\nabla \mathcal{L}_{\mathrm{NAR}})$; and (3) adaptive weighting $\alpha_k \propto \exp(-\|\nabla_{s_k} \mathcal{L}\|/\sigma_t)$. NAR only competes when sessions are independent or under strict latency constraints.*

This theoretically-grounded approach proves particularly effective for modeling multi-guidance user interest evolution, where the AR structure's ability to decompose complex joint distributions into conditional chains captures interest shifts more accurately than NAR's mean-field approximation. This advantage that grows superlinearly with session count $K$ (Figure 4b).

### 4.3 IMPROVING EFFICIENCY

To meet the low-latency requirements of online CTR serving, we optimize the diffusion-based interest fusion via consistency constraints. Traditional diffusion models suffer from slow inference due to iterative denoising steps, which is prohibitive for real-time applications. We introduce a consistency loss that enforces similarity between interest representations generated under different noise levels:

$$\mathcal{L}_{cons} = \mathbb{E}_{t_1, t_2 \sim p(t)} \left[ \|f_\theta(\mathbf{x}_{t_1}, t_1) - f_\theta(\mathbf{x}_{t_2}, t_2)\|^2 \right] \tag{15}$$

This constraint enables high-quality generation with drastically fewer sampling steps by learning noise-invariant representations. This makes diffusion models practical for industrial CTR systems while preserving their expressive power.

### 4.4 THEORETICAL ANALYSIS UNDER ZERO-DATA SCENARIOS

We analyze the zero-data scenario through the lens of denoising diffusion theory. The key insight is that diffusion models provide a principled mechanism for navigating the interest manifold $\mathcal{M}$ even with completely uninformative inputs.

**Theorem 3 (Diffusion Manifold Consistency)** *Under mild smoothness assumptions on the interest manifold $\mathcal{M}$, the diffusion process generates representations that remain close to $\mathcal{M}$:*

$$d_{\mathcal{M}}(z_0, z^*) \leq C \cdot \mathbb{E}[\|\epsilon - \epsilon_\theta(z_t, t | z_l, z_s)\|] + \epsilon_{approx} \tag{16}$$

Theorem 3 establishes that the learned score function guides the reverse diffusion process toward semantically meaningful regions of the representation space. This geometric constraint ensures robustness even when behavioral data is completely absent.

**Theorem 4 (Zero-Data Denoising Optimality)** *When $z_l, z_s$ are uninformative, the optimal denoising strategy converges to sampling from population-level statistics:*

$$\epsilon_\theta(z_t, t) = \mathbb{E}_{z_0 \sim p_{data}}[\epsilon | z_t] \tag{17}$$

This result demonstrates that our diffusion-based framework naturally falls back to reasonable population-level priors in zero-data scenarios. Detailed derivations and experiment are provided in Appendix M and N.

## 4.5 OVERALL TRAINING AND INFERENCE PROCESS

The overall training objective combines four loss components:

$$\mathcal{L} = \underbrace{\mathcal{L}_{\text{CE}}}_{\text{CTR prediction}} + \underbrace{\lambda_1 \mathcal{L}_{\text{Evol}}}_{\text{evolutionary}} + \underbrace{\lambda_2 \mathcal{L}_{\text{Dist}}}_{\text{disentanglement}} + \underbrace{\lambda_3 \mathcal{L}_{\text{cons}}}_{\text{consistency}} + \beta \|\Theta\|_2 \tag{18}$$

where $\mathcal{L}_{\text{CE}}$ is the cross-entropy loss, $\mathcal{L}_{\text{Evol}} = \frac{1}{N} \sum_{i=1}^{N} D_{\cos}(\mathbf{s}_{i,k+1}^{\text{pred}}, \mathbf{s}_{i,k+1}^{\text{true}})$ captures interest evolution, $\mathcal{L}_{\text{Dist}} = \|\mathbf{g}_{\text{core}}^{\top} \mathbf{g}_{\text{fluct}}\|_2^2$ enforces guidance disentanglement, and $\mathcal{L}_{\text{cons}}$ for improving efficiency. The training protocol of **iFusion** is shown in Algorithm 1. During inference, we employ an iterative denoising process starting from $\mathbf{x}_T = \mathbf{h}^{\text{L}}$. At each step $t$ from $T$ to 1, we compute:

$$\mathbf{x}_{t-1} = \frac{\sqrt{\bar{\alpha}_{t-1}}\beta_t}{1 - \bar{\alpha}_t} \tilde{f}_\theta(\mathbf{x}_t, \{\mathbf{h}_i^{\text{S}}\}_{i=1}^{K}, t) + \frac{\sqrt{\alpha_t}(1 - \bar{\alpha}_{t-1})}{1 - \bar{\alpha}_t} \mathbf{x}_t + \sqrt{\tilde{\beta}_t} \mathbf{z} \tag{19}$$

This formulation represents a single denoising step, which is applied iteratively until reaching the final fused representation $\mathbf{x}_0$ for CTR prediction. The inference protocol of **iFusion** is show in Algorithm 2.

## 5 EXPERIMENTS

In this section, we present a comprehensive empirical evaluation of our proposed method. We mainly answer the following research questions: **RQ1**: How does iFusion perform compared with other user behavior modeling methods? **RQ2**: How do different components of iFusion benefit its performance? **RQ3**: What is the impact of factors (e.g., sample steps) on iFusion's performance? **RQ4**: How efficient is iFusion inference?

### 5.1 EXPERIMENTAL SETTINGS

**Datasets and Evalution Metrics.** We conducted experiments on benchmark including Amazon Book Dataset, Taobao Dataset and Ali Ads Dataset which are widely used in user behavior modeling research, as well as a real industrial dataset from a large internet advertising platform. A detailed description of the datasets can be found in Appendix G. We use AUC as the evaluation metric for offline experiments. Furthermore, we adopt the relative improvement (RelaImpr) metric to evaluate the performance difference between models (Zhou et al., 2018; Yan et al., 2014).

$$\text{RelaImpr} = \left( \frac{\text{AUC(model)} - 0.5}{\text{AUC(base model)} - 0.5} - 1 \right) \times 100\% \tag{20}$$

**Baselines.** We compare to mainstream user behavior modeling algorithms including Avg-Pooling DNN, DIN (Zhou et al., 2018), DIEN (Zhou et al., 2019), SIM (Pi et al., 2020), ETA (Chen et al., 2021), SDIM (Cao et al., 2022), TWIN (Chang et al., 2023), TWIN-V2 (Si et al., 2024), MTGR (Han et al., 2025), DiffuRec (Li et al., 2023), DreamRec (Yang et al., 2023) and DiffuMIN (Lai et al., 2025). Detailed descriptions of each baseline and implementation are provided in Appendix H.1. All models were implemented using TensorFlow. For model training, we used Adam as the optimizer and trained each model for a single epoch. More details can be found in Appendix H.2.

### 5.2 COMPARISON WITH BASELINES (RQ1)

We evaluate iFusion against state-of-the-art baselines in Table 1 and provide the statistical significance of our model's improvement over the best baseline model. Notably, in CTR prediction scenarios, even a 0.001 AUC gain is considered practically significant (Zhou et al., 2018; 2019). Our experimental analysis reveals several key observations. First, both DIN and DIEN demonstrate superior performance over the standard DNN, validating the importance of attention mechanisms in modeling user behavior sequences. Furthermore, specialized architectures for long-term behavior modeling exhibit enhanced capability in capturing extended user interest patterns. Methods like

---

http://jmcauley.ucsd.edu/data/amazon/
https://tianchi.aliyun.com/dataset/649
https://tianchi.aliyun.com/dataset/56

Table 1: Performance comparison on four datasets. Best results are in **bold**. All results run over 3 times with std $\approx$ 1e-3

| Method | Amazon | | Taobao | | Ali Ads | | Industrial | |
|---|---|---|---|---|---|---|---|---|
| | AUC | RelaImpr | AUC | RelaImpr | AUC | RelaImpr | AUC | RelaImpr |
| *Traditional Methods* | | | | | | | | |
| AvgPooling DNN | 0.7689 | 0.00% | 0.8539 | 0.00% | 0.6352 | 0.00% | 0.7512 | 0.00% |
| DIN | 0.8162 | +17.59% | 0.8995 | +12.88% | 0.6422 | +5.18% | 0.7564 | +2.07% |
| DIEN | 0.8377 | +25.69% | 0.9222 | +19.30% | 0.6431 | +5.84% | 0.7611 | +3.94% |
| SIM | 0.8420 | +27.18% | 0.9268 | +20.60% | 0.6587 | +17.38% | 0.7625 | +4.50% |
| ETA | 0.8422 | +27.26% | 0.9272 | +20.71% | 0.6591 | +17.68% | 0.7625 | +4.50% |
| SDIM | 0.8426 | +27.41% | 0.9277 | +20.85% | 0.6596 | +18.05% | 0.7628 | +4.62% |
| TWIN | 0.8431 | +27.59% | 0.9288 | +21.16% | 0.6601 | +18.42% | 0.7630 | +4.70% |
| TWIN-V2 | 0.8433 | +27.67% | 0.9289 | +21.19% | 0.6607 | +18.86% | 0.7634 | +4.86% |
| MTGR | 0.8440 | +27.93% | 0.9296 | +21.39% | 0.6615 | +19.45% | 0.7648 | +5.41% |
| *Diffusion-based Generative Methods* | | | | | | | | |
| DiffuRec$_{ctr}$ | 0.8395 | +26.26% | 0.9258 | +20.32% | 0.6584 | +17.16% | 0.7607 | +3.78% |
| DreamRec$_{ctr}$ | 0.8421 | +27.22% | 0.9286 | +21.11% | 0.6590 | +17.60% | 0.7619 | +4.26% |
| DiffuMIN | 0.8427 | +27.45% | 0.9288 | +21.16% | 0.6595 | +17.97% | 0.7623 | +4.42% |
| **iFusion (Ours)** | **0.8512** | **+30.61%** | **0.9347** | **+22.83%** | **0.6652** | **+22.19%** | **0.7685** | **+6.89%** |

MTGR achieve additional gains through direct modeling of complete user behavior chains. Notably, existing diffusion-based approaches face limitations in their guidance mechanisms, struggling to disentangle core user preferences from transient fluctuations during interest representation learning. This entanglement issue hinders their performance in CTR prediction tasks. In contrast, iFusion addresses this fundamental challenge through interest decoupling guidance and autoregressive denoising generation that progressively refines interest modeling. This principled approach yields statistically significant improvements over all competitive baselines, demonstrating the effectiveness of our proposed framework.

## 5.3 ABLATION STUDY (RQ2)

To systematically evaluate the contribution of each component in iFusion, we conduct comprehensive ablation studies. The results, summarized in Figure 3, provide compelling evidence for our design choices. In Figure 3a, our proposed DCFG demonstrates significant advantages over conventional CFG approaches. By separating guidance signals for core preferences and transient fluctuations, DCFG achieves more precise interest representation, as reflected in the consistent AUC improvements. We observe that naively incorporating all guidance information leads to degraded performance in interest fusion quality. This finding underscores the necessity of our carefully designed guidance modulation strategy in interest fusion task, which selectively integrates relevant signals while filtering out noise. In Figure 3b, the introduction of MARN brings substantial gains, attributed to its hierarchical processing of interest guidance across temporal sessions. Increasing the complexity of MARN's internal networks does not yield significant improvements, as we are fusing in interest space rather than original sequence. It also suggest that the effectiveness stems from our interest-space fusion paradigm rather than network capacity. As shown in Figure 3c, the consistency loss enables remarkable acceleration in inference generation while maintaining performance. This demonstrates the practical viability of iFusion for real-time industrial deployment.

## 5.4 HYPER-PARAMETER STUDY (RQ3)

We conduct a systematic analysis of iFusion's sensitivity to key hyper parameters: noise scheduling strategy, inference sampling steps, and the number of behavioral sessions. Our findings reveal important insights into the model's operational characteristics. Figure 4a demonstrates that under consistency constraints, various noise schedules achieve peak performance with minimal sampling steps. The cosine schedule emerges as optimal, achieving best AUC with just a single inference step. Notably, performance degrades with increased sampling steps, likely due to error accumulation in the iterative generation process. This observation validates our design choice for efficient few-step inference. As shown in Figure 4b, the advantage of MARN's autoregressive processing becomes

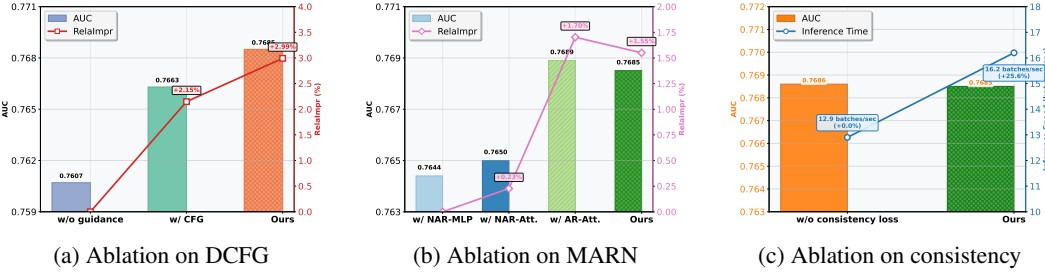

(a) Ablation on DCFG  (b) Ablation on MARN  (c) Ablation on consistency

Figure 3: Ablation study on industrial dataset over 3 runs (std ≈ 1e-3).

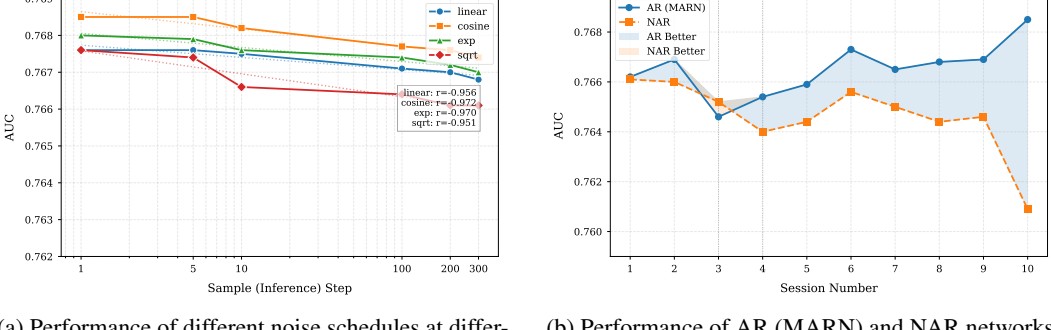

(a) Performance of different noise schedules at different sample steps training with consistency loss

(b) Performance of AR (MARN) and NAR networks under different number of sessions

Figure 4: Hyperparameter study on industrial dataset over 3 runs (std ≈ 1e-3)

increasingly pronounced with more sessions, confirming our theoretical analysis in Appendix E. This scalability demonstrates iFusion's suitability for modeling complex interest fusion. Based on our analysis, we adopt cosine noise scheduling with single-step inference as the default configuration, achieving an optimal balance between performance and computational efficiency. Additional hyperparameter studies are provided in Appendix I due to space constraints.

## 5.5 EFFICIENCY ANALYSIS (RQ4)

We conduct a comprehensive efficiency evaluation of iFusion across both offline and online deployment scenarios. The results demonstrate the practical viability of our approach for industrial applications. In offline inference, by integrating our interest fusion module atop the optimal baseline model, we observe only a marginal 0.3% increase in inference time cost. This negligible overhead confirms the computational efficiency of our architectural design. In production environments, iFusion introduces merely a 0.302% increase in TP99 latency compared to a highly-optimized industrial-grade model. Such minimal performance impact is well within acceptable thresholds for large-scale deployment. These efficiency metrics, coupled with the significant performance gains demonstrated in previous sections, establish iFusion as an effective and practical solution for real-world CTR prediction systems. Additional efficiency analyses, including training time convergence and memory utilization patterns, are provided in Appendix J due to space constraints.

## 5.6 ONLINE A/B TESTS

We conducted large-scale online A/B tests over a 7-day period to evaluate iFusion's real-world performance in a production advertising system. The experiment involved hundreds of millions of users, with iFusion demonstrating statistically significant improvements across key business metrics: achieving a +2.44% gain in CTR ($p\_value < 0.001$) and a +2.61% gain in eCPM ($p\_value < 0.001$). These results confirm iFusion's effectiveness in simultaneously enhancing user engagement and platform monetization, while maintaining acceptable computational overhead as detailed in Section 5.5.

## 6 CONCLUSIONS

We propose iFusion, a diffusion-based generative framework for interest fusion that overcomes key limitations of existing methods: misaligned heterogeneous feature spaces, disjointed modeling in late-fusion paradigms, and perturbation-interest entanglement. By reformulating interest fusion as a conditional generation process, iFusion integrates long-term and short-term interest through decoupled denoising without restrictive linear assumptions. The framework incorporates two novel components: the MARN architecture, which jointly models interest evolution and fusion via autoregressive dynamics, and the DCFG mechanism, which disentangles core preferences from transient fluctuations to enhance robustness. Extensive evaluations in CTR prediction demonstrate the effectiveness of our approach. This work advances user interest fusion under a diffusion paradigm, enabling fluctuation-aware interest modeling. The framework is readily transferable to other behavioral interest fusion tasks such as CVR and GMV prediction.

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

## A THE USE OF LARGE LANGUAGE MODELS (LLMS)

In the preparation of this paper, we employed a large language model primarily for language polishing and expression refinement. The LLM was used as an assistive tool to enhance the clarity, coherence, and fluency of the writing.

## B NOTATIONS

Table 2: Notation Summary

| Symbol | Description |
|---|---|
| $\mathcal{B}^{\mathrm{L}}$ | long-term behavior sequence |
| $b_i^{\mathrm{L}}$ | the i-th user's long-term behavior |
| $M$ | number of long-term user behaviors |
| $\mathcal{B}^{\mathrm{S}}$ | short-term behavior sequence |
| $S_i$ | short-term behavior sequence |
| $b_i^{\mathrm{S}}$ | the i-th user's short-term behavior session |
| $K$ | number of short-term user behaviors |
| $f_{\mathrm{L}}(\cdot)$ | long-term behavior encoder |
| $f_{\mathrm{S}}(\cdot)$ | short-term session encoder |
| $\mathbf{h}^{\mathrm{L}}$ | long-term interest representation |
| $\mathbf{h}_i^{\mathrm{S}}$ | short-term session interest representation |
| $\mathbf{h}^{\mathrm{fusion}}$ | generated fused interest representation |
| $\mathcal{F}_\theta$ | user interest fusion function |
| $T$ | diffusion steps |
| $\beta_t$ | variance schedule for diffusion model |
| $\mathbf{x}_t$ | the noisy representation at time step $t$ |
| $\hat{\mathbf{x}}_t$ | the noisy representation at time step $t$ in the reverse process, for simplicity, we define $\hat{\mathbf{x}}_T = \mathbf{x}_T$ |
| $g_{cp}$ | core preference guidance |
| $g_{tf}$ | transient fluctuation guidance |
| $\gamma_j, j \in \{cp, tf\}$ | hyper-parameter to control the strength of core preference guidance and transient fluctuation guidance |

## C DERIVATION OF EQUATION 14 VIA BAYESIAN GRADIENT DECOMPOSITION

Starting from the joint conditional distribution:

$$p(\mathbf{x}_t \mid g_{cp}, g_{tf}) = \frac{p(g_{cp}, g_{tf} \mid \mathbf{x}_t)p(\mathbf{x}_t)}{p(g_{cp}, g_{tf})} \tag{21}$$

$$\Rightarrow \log p(\mathbf{x}_t \mid g_{cp}, g_{tf}) = \log p(g_{cp}, g_{tf} \mid \mathbf{x}_t) + \log p(\mathbf{x}_t) + \mathrm{const.}$$

Taking the gradient with respect to $x_t$:

$$\nabla_{\mathbf{x}_t} \log p(\mathbf{x}_t \mid g_{cp}, g_{tf}) = \nabla_{\mathbf{x}_t} \log p(g_{cp}, g_{tf} \mid \mathbf{x}_t) + \nabla_{\mathbf{x}_t} \log p(\mathbf{x}_t) \tag{22}$$

Under conditional independence:

$$\nabla_{\mathbf{x}_t} \log p(g_{cp}, g_{tf} \mid \mathbf{x}_t) = \sum_{i \in \{cp, tf\}} \nabla_{\mathbf{x}_t} \log p(g_i \mid \mathbf{x}_t) \tag{23}$$

Reorganizing terms using the identity $\nabla_{\mathbf{x}_t} \log p(g_i \mid \mathbf{x}_t) = \nabla_{\mathbf{x}_t} \log p(\mathbf{x}_t \mid g_i) - \nabla_{\mathbf{x}_t} \log p(\mathbf{x}_t)$ yields the final form:

$$\nabla_{\mathbf{x}_t} \log p(\mathbf{x}_t \mid g_{cp}, g_{tf}) = (1 - \gamma_{cp} - \gamma_{tf})\nabla_{\mathbf{x}_t} \log p(\mathbf{x}_t) + \gamma_{cp}\nabla_{\mathbf{x}_t} \log p(\mathbf{x}_t \mid g_{cp})$$
$$+ \gamma_{tf}\nabla_{\mathbf{x}_t} \log p(x_t \mid g_{tf}) \tag{24}$$

Equivalent to:

$$\nabla_{\mathbf{x}_t} \log p(\mathbf{x}_t \mid g_{cp}, g_{tf}) = \nabla_{\mathbf{x}_t} \log p(\mathbf{x}_t) + \gamma_{cp}(\nabla_{\mathbf{x}_t} \log p(\mathbf{x}_t \mid g_{cp}) - \nabla_{\mathbf{x}_t} \log p(\mathbf{x}_t))$$
$$+ \gamma_{tf}(\nabla_{\mathbf{x}_t} \log p(\mathbf{x}_t \mid g_{tf}) - \nabla_{\mathbf{x}_t} \log p(\mathbf{x}_t)) \tag{25}$$

# D    PROOF OF POTENTIAL DECOMPOSITION IN LANGEVIN DYNAMICS

## D.1    PRELIMINARIES

Consider a 2D overdamped Langevin system with position $\mathbf{r} = (x, y)$:

$$\gamma \frac{d\mathbf{r}}{dt} = -\nabla V(\mathbf{r}) + \sqrt{2\gamma k_B T} \boldsymbol{\eta}(t) \tag{26}$$

where $\boldsymbol{\eta}(t) = (\eta_x(t), \eta_y(t))$ is standard white noise satisfying:

$$\langle \eta_i(t) \rangle = 0$$
$$\langle \eta_i(t) \eta_j(t') \rangle = \delta_{ij} \delta(t - t')$$

**Assumption 1 (Potential Separability)** *The potential $V(x, y)$ admits the decomposition:*

$$V(x, y) = V_x(x) + V_y(y) + \epsilon V_{xy}(x, y) \tag{27}$$

*where $\epsilon = 0$ for the exactly separable case.*

## D.2    EXACT DECOMPOSITION THEOREM

**Theorem 5 (Dynamics Decoupling)** *For $\epsilon = 0$ in equation 27, the system equation 26 decouples exactly into:*

$$\gamma \dot{x} = -\partial_x V_x(x) + \sqrt{2\gamma k_B T} \eta_x(t) \tag{28}$$

$$\gamma \dot{y} = -\partial_y V_y(y) + \sqrt{2\gamma k_B T} \eta_y(t) \tag{29}$$

**Proof 1** *The gradient operator acts on equation 27 as:*

$$\nabla V = \begin{pmatrix} \partial_x V_x(x) + \epsilon \partial_x V_{xy}(x, y) \\ \partial_y V_y(y) + \epsilon \partial_y V_{xy}(x, y) \end{pmatrix} \tag{30}$$

*For $\epsilon = 0$, substituting into equation 26 yields immediate decoupling into equation 28 and equation 29. The noise terms remain uncorrelated since:*

$$\langle \eta_x(t) \eta_y(t') \rangle = 0 \quad \forall t, t' \tag{31}$$

## D.3    PROBABILITY DENSITY FACTORIZATION

**Lemma 1 (FP Equation Decomposition)** *The Fokker-Planck equation for equation 26 with $\epsilon = 0$:*

$$\frac{\partial P}{\partial t} = \nabla \cdot \left[ \frac{1}{\gamma} (\nabla V) P + D \nabla P \right], \quad D = \frac{k_B T}{\gamma} \tag{32}$$

*admits separable solutions $P(x, y, t) = P_x(x, t) P_y(y, t)$.*

**Proof 2** *Substitute the separable ansatz into equation 32:*

$$\frac{\partial}{\partial t}(P_x P_y) = \frac{1}{\gamma} \left[ \frac{\partial}{\partial x}(\partial_x V_x P_x P_y) + \frac{\partial}{\partial y}(\partial_y V_y P_x P_y) \right]$$
$$+ D \left[ \frac{\partial^2}{\partial x^2}(P_x P_y) + \frac{\partial^2}{\partial y^2}(P_x P_y) \right]$$

*Expanding derivatives and dividing through by $P_x P_y$:*

$$\frac{1}{P_x} \frac{\partial P_x}{\partial t} + \frac{1}{P_y} \frac{\partial P_y}{\partial t} = \frac{1}{\gamma} \left[ \frac{1}{P_x} \frac{\partial}{\partial x}(\partial_x V_x P_x) + \frac{1}{P_y} \frac{\partial}{\partial y}(\partial_y V_y P_y) \right] + D \left[ \frac{1}{P_x} \frac{\partial^2 P_x}{\partial x^2} + \frac{1}{P_y} \frac{\partial^2 P_y}{\partial y^2} \right] \tag{33}$$

*This separates into two independent equations:*

$$\frac{\partial P_x}{\partial t} = \frac{1}{\gamma} \frac{\partial}{\partial x}(\partial_x V_x P_x) + D \frac{\partial^2 P_x}{\partial x^2} \tag{34}$$

$$\frac{\partial P_y}{\partial t} = \frac{1}{\gamma} \frac{\partial}{\partial y}(\partial_y V_y P_y) + D \frac{\partial^2 P_y}{\partial y^2} \tag{35}$$

*confirming the solution's separability.*

### D.4 PERTURBATION ANALYSIS FOR WEAK COUPLING

For $\epsilon \ll 1$, we can expand the solution as:

$$P(x, y, t) = P_x^{(0)}(x, t)P_y^{(0)}(y, t) + \epsilon P^{(1)}(x, y, t) + O(\epsilon^2) \tag{36}$$

where the superscript $(0)$ denotes the separable solution. The first-order correction satisfies:

$$\frac{\partial P^{(1)}}{\partial t} = \mathscr{L}_x P^{(1)} + \mathscr{L}_y P^{(1)} + \frac{1}{\gamma} \nabla \cdot (P_x^{(0)} P_y^{(0)} \nabla V_{xy}) \tag{37}$$

with $\mathscr{L}_i$ being the FP operators for each coordinate. This shows how non-separable terms introduce coupling between directions.

## E   THEORETICAL ANALYSIS OF AR VS. NAR IN MULTI-GUIDANCE INJECTION IN DIFFUSION REVERSE PROCESS

The core challenge lies in effectively integrating $K$ short-term behavior sessions $\mathbf{s}_{1:K}$ into the diffusion reverse process. We analyze two fundamental approaches through three lenses: **Modeling Perspective.** When sessions exhibit weak dependence ($I(s_i; s_j) \leq \epsilon$), the autoregressive (AR) injection's chain rule decomposition achieves strictly lower approximation error. For any diffusion step $t$:

**Theorem 6 (Approximation Error Bound)** *The KL-divergence gap between AR and non-autoregressive (NAR) injection satisfies:*

$$D_{KL}^{NAR} - D_{KL}^{AR} \geq \underbrace{\frac{1}{2} \sum_{k=1}^{K} (\alpha_k - \frac{1}{K})^2}_{\text{Weight mismatch}} - \underbrace{\sum_{i<j} I(s_i; s_j)}_{\text{Correlation penalty}} \tag{38}$$

**Proof 3** *The AR model's sequential processing preserves conditional dependencies through exact probability chain rule:*

$$p_{AR}(\mathbf{x}_{t-1}|\mathbf{s}_{1:K}) = \prod_{k=1}^{K} p(\mathbf{x}_{t-1}|s_k, \mathbf{x}_t, s_{1:k-1}) \tag{39}$$

*whereas NAR's concatenation forces mean-field approximation, introducing the weight mismatch term. The correlation penalty emerges from Jensen's inequality applied to the joint distribution.*

**Optimization Dynamics.** The gradient behavior differs markedly due to injection architecture. AR's sequential nature induces implicit gradient averaging:

**Theorem 7 (Gradient Variance Ratio)** *For $L$-Lipschitz denoising networks with session weights $\alpha_k$:*

$$\frac{Var(\nabla_\theta \mathcal{L}_{NAR})}{Var(\nabla_\theta \mathcal{L}_{AR})} \geq \frac{K}{L^2} \cdot \frac{\|\mathbf{J}_{NAR}\|_F^2}{\sum_{k=1}^{K} \alpha_k^2} \tag{40}$$

*where $\mathbf{J}$ denotes the input Jacobian matrix.*

**Proof 4** *AR's variance scales as $O(1/K)$ due to Central Limit Theorem effects in sequential processing, while NAR's variance grows with input dimension $d_{in}$:*

$$Var_{NAR} \propto \mathbb{E}[\|Concat(s_1, ..., s_K)\|^2] \approx K \cdot d_{in} \tag{41}$$

*The ratio follows from direct variance computation and Lipschitz continuity.*

**Adaptive Weighting.** AR models automatically learn noise-sensitive session weights without explicit design:

**Proposition 1 (Implicit Attention)** *At noise level $\sigma_t$, AR's effective weights satisfy:*

$$\alpha_k^{AR}(t) \propto \exp\left(-\frac{\mathbb{E}[\|\nabla_{s_k}\mathcal{L}\|]}{\sigma_t}\right) \tag{42}$$

**Proof 5** *Through backpropagation, the gradient norms $\|\nabla_{s_k}\mathcal{L}\|$ act as implicit importance scores. Higher noise levels $\sigma_t$ soften the weight distribution, matching the intuition that session relevance becomes ambiguous in early diffusion steps.*

Table 3: Practical Implications Summary

| Property | Interpretation |
|---|---|
| Lower KL-bound | AR better preserves complex session interactions |
| Gradient stability | AR's SNR improves with more sessions ($\propto \sqrt{K}$) |
| Dynamic weighting | AR adapts to session relevance without architecture changes |

The theoretical findings suggest AR injection is preferable when: (1) session correlations exist ($\epsilon > 0$), (2) model capacity permits sequential processing, and (3) gradient stability is critical. NAR remains competitive when sessions are truly independent or computational latency dominates quality concerns.

## F  PSEUDOCODE FOR IFUSION TRAINING AND INFERENCE

---
**Algorithm 1** iFusion Training Protocol

---
**Require:** Training set $\mathcal{D}$, core signal extractor $g_1$, fluctuation extractor $g_2$
**Require:** Dropout rates $p_{\text{drop}}^1, p_{\text{drop}}^2$ Initialize all model parameters.
 1: **repeat**
 2:     $(\mathbf{h}^L, \mathbf{h}_{1:K+1}^S) \sim \mathcal{D}$                       ▷ Get long-term and short-term behavior interests.
 3:     $\mathbf{x}_0 = \mathbf{h}^L$
 4:     $t_1, t_2 \sim \mathcal{U}(\{1,\ldots,T\})$                       ▷ Sample diffusion step.
 5:     $\epsilon_1, \epsilon_2 \sim \mathcal{N}(0, I)$                       ▷ Sample Gaussian noise.
 6:     **for** each $\{t, \epsilon\} \in (\{t_1, \epsilon_1\}, \{t_2, \epsilon_2\})$ do following step:
 7:         $\mathbf{x}_t = \sqrt{\alpha_t}\mathbf{x}_0 + \sqrt{1-\alpha_t}\epsilon$                       ▷ Forward process.
 8:         $g_{cp}^{1:K}, g_{tf}^{1:K} = e_1(\mathbf{h}_{1:K}), e_2(\mathbf{h}_{1:K})$                       ▷ Disentangle guidance signals
 9:         $m_i \sim \text{Bern}(1 - p_{\text{drop}}^i)$                       ▷ Perform unconditional training with probability $p_{\text{drop}}$
10:         **for** $mode$ in $\{\text{evo}, \text{task}\}$ **do**
11:             If $mode = \text{evo}$ then $idx = 1:K$ else $idx = 2:K+1$                       ▷ Select used sessions
12:             $\hat{g}_{cp}^{mode} = m_1 \cdot g_{cp}^{idx} + (1-m_1) \cdot \emptyset$
13:             $\hat{g}_{tf}^{mode} = m_2 \cdot g_{tf}^{idx} + (1-m_2) \cdot \emptyset$
14:             $\hat{f}_\theta^{mode} = \frac{\gamma_{cp}}{\gamma_{cp}+\gamma_{tf}} f_\theta(\mathbf{x}_t, t, \hat{g}_{cp}^{mode}) + \frac{\gamma_{tf}}{\gamma_{cp}+\gamma_{tf}} f_\theta(\mathbf{x}_t, t, \hat{g}_{tf}^{mode})$                       ▷ equation 4
15:         **end for**
16:         $\mathcal{L} = \mathcal{L}_{\text{CE}} + \lambda_1 \mathcal{L}_{\text{Evol}} + \lambda_2 \mathcal{L}_{\text{Dist}} + \lambda_3 \mathcal{L}_{\text{cons}}$                       ▷ equation 18
17:         Update $\theta \leftarrow \theta - \eta \nabla_\theta \mathcal{L}$
18: **until** convergence

---

---

**Algorithm 2** iFusion Inference Protocol

---

**Require:** Trained model $f_\theta$, trained core signal extractor $g_1$, trained fluctuation extractor $g_2$
**Require:** long-term interest $\mathbf{h}^L$, short-term sessions $\mathbf{h}^S_{1:K}$
**Ensure:** generated interest $\mathbf{x}_0$
 1: Initialize $\mathbf{x}_T$ with long-term interest $\mathbf{h}^L$
 2: $g^{1:K}_{cp}, g^{1:K}_{tf} = e_1(\mathbf{h}^S_{1:K}), e_2(\mathbf{h}^S_{1:K})$
 3: **for** $t = T$ **down to** 1 **do**
 4:     $\mathbf{z} \sim \mathcal{N}(0, I)$ if $t > 1$ else $\mathbf{z} = 0$
 5:     $f_{\text{uncond}} = f_\theta(\mathbf{x}_t, t, \emptyset)$
 6:     $f_{\text{cp}} = f_\theta(\mathbf{x}_t, t, (g^{1:K}_{cp}))$
 7:     $f_{\text{tf}} = f_\theta(\mathbf{x}_t, t, (g^{1:K}_{tf}))$
 8:     $\hat{f} = f_{\text{uncond}} + \sum_{j \in \{cp, tf\}} \gamma_j (f_j - f_{\text{uncond}})$                    ▷ Theorem 14
 9:     $\mathbf{x}_{t-1} = \frac{\sqrt{\bar{\alpha}_{t-1}}\beta_t}{1-\bar{\alpha}_t}\hat{f} + \frac{\sqrt{\bar{\alpha}_t}(1-\bar{\alpha}_{t-1})}{1-\bar{\alpha}_t}\mathbf{x}_t + \sqrt{\tilde{\beta}_t}\mathbf{z}$
10: **end for**

---

# G    DETAILS OF DATASET

To better demonstrate our method's capability in addressing the challenges presented in this paper, we have made several deliberate and distinctive choices in constructing the dataset, particularly regarding the public dataset. Below, we provide a detailed overview of the dataset.

**Amazon Dataset** (McAuley et al., 2015) is a commonly used benchmark in user behavior modeling (Zhou et al., 2018; 2019). It comprises product reviews and metadata from Amazon, specifically utilizing the Books subset, which includes 75,053 users, 358,367 items, and 1,583 categories. In this dataset, reviews are treated as interaction behaviors and are chronologically sorted per user, with a maximum behavior sequence length of 100. Following common practices in related work, we split the sequence into short-term and long-term sequential features, using the most recent 10 interactions for short-term modeling and the preceding 90 for long-term representation.

**Taobao Dataset** contains approximately one million randomly sampled users, recording all their interactions (e.g., clicks, purchases, add-to-cart, and likes) between November 25 and December 3, 2017. The dataset includes user IDs, item IDs, item category IDs, behavior types, and timestamps. Following (Chen et al., 2021), we first sorted each user's behaviors chronologically. Unlike previous works that rely solely on click data, we intentionally incorporate all behavior types when constructing the short-term behavior sequence. This design choice allows us to rigorously evaluate our method's capability in addressing the first challenge: misaligned heterogeneous feature spaces arising from divergent feature selection mechanisms, as different behavior types naturally introduce variations in feature distributions and semantics. For the short-term behavior sequence, we extracted the most recent 16 behaviors. Meanwhile, the long-term behavior sequence was constructed directly from the user's most recent 256 behaviors.

**Alibaba Ads Dataset** is provided by Alibaba, which is a display advertising click-through rate prediction dataset. It includes shopping behavior data from all users over a 22-day period and includes comprehensive information on users, advertisements, and user behaviors.

**Industrial Dataset** is the traffic logs from the advertising system of a large internet platform. 10 billion click-through logs in the first 32 days are used for training, and 0.5 million from the following day for testing. We define the user's click behavior from the past 180 days as the long-term behavior sequence, while the interactions from the most recent 60 days form the short-term behavior sequence. Notably, this short-term sequence represents a heterogeneous mixture of search queries, item clicks, and purchase actions, which directly exemplifies the challenge of misaligned heterogeneous feature spaces arising from divergent feature selection mechanisms as outlined in our introduction. Furthermore, the short-term sequence is segmented into sessions based on predefined rules to better capture temporal patterns and user intent transitions within these behaviorally diverse interactions.

# H  DETAILS OF EXPERIMENT SETTINGS

## H.1  DETAILS OF BASELINES AND IMPLEMENTATION

We compare to mainstream user behavior modeling algorithms including:

- **Avg-Pooling DNN**: All user behaviors are treated equally with the sum pooling operation.
- **DIN**: DIN introduces an attention mechanism to capture user interests by adaptively weighting the historical behaviors based on their relevance to the target item.
- **DIEN**: DIEN models the dynamic evolution of user interests over time using a GRU-based network with an auxiliary loss to enhance behavior representation learning.
- **SIM**: SIM extracts user interests with two cascaded search units: (i) General Search Unit (GSU) acts as a general search from the raw and arbitrary long sequential behavior data, with query information from candidate item, and gets a Sub user Behavior Sequence (SBS) which is relevant to candidate item; (ii) Exact Search Unit (ESU) models the precise relationship between candidate item and SBS.
- **ETA**: Compared with SIM, the most important improvement of ETA is the change of the retrieval method in the GSU stage, which changes the simple category-based retrieval of SIM to the retrieval based on Hamming distance after the item embedding is processed by SimHash.
- **SDIM**: SDIM sums the embeddings of items with the same hash as the target item in the user behavior sequence and normalizes them to obtain the user interest expression, further reducing the time complexity to $O\left(Lmlog\left(d\right)\right)$. $B$ represents the number of candidate sets, $L$ represents the sequence length, $d$ represents the item embedding dimension, $K$ represents the top K items selected, and $m$ represents the number of different hash functions used in SDIM.
- **TWIN**: TWIN is a two-stage interest network for lifelong user behavior modeling, in which Consistency-Preserved GSU (CP-GSU) adopts the same target behavior correlation measure as TA in ESU, making the two stages twins.
- **TWIN V2**: TWIN-V2 compresses lifecycle behaviors through clustering and discovers more accurate and diverse user interests. In the offline phase, a hierarchical clustering method groups items with similar characteristics in lifecycle behaviors into clusters. By limiting the size of clusters, behavior sequences can be compressed to facilitate online reasoning in GSU retrieval. Cluster-aware target attention extracts users' long-term interests, resulting in more accurate and diverse recommendations.
- **MTGR**: MTGR integrates the advanced DLRM and GRM modes, retaining the characteristics of DLRM such as the Cross feature, while verifying the excellent performance of GRM and proposing Group-Layer Norm and dynamic masking strategies.
- **DiffuRec**: DiffuRec is the first diffusion-based sequential recommendation model that constructs dynamic item representations through controlled noise injection, enhancing uncertainty modeling in user preference learning.
- **DreamRec**: DreamRec is a diffusion-based sequential recommendation method that learns to generate target items by denoising with guidance from historical interactions using classifier-free guidance mechanism, eliminating the need for negative sampling while directly capturing user preferences. In our implementation, we use long-term behavior as the starting point of the forward process and use the fusion representation of short-term behaviors from multiple sessions as guidance.
- **DiffuMIN**: DiffuMIN is designed to model long-term user behavior and deeply explore the user interest space. It uses a goal-oriented multi-interest extraction method that first performs an orthogonal decomposition of the goal to obtain interest channels. It then decouples and extracts multiple user interests by modeling the relationship between interest channels and user behavior. A diffusion module guided by contextual interests and interest channels is then employed. This module anchors the user's personalized and goal-oriented interest types, generating enhanced interests that are consistent with the user's latent interest space, further exploring the restricted interest space. Finally, contrastive learning is utilized to ensure that the generated enhanced interests are consistent with the user's true preferences.

## H.2 MORE DETAILS OF IFUSION IMPLEMENTATION

Our long-term encoder employs a double-layer transformer encoder-decoder, while the short-term encoder uses a single-layer transformer encoder to process each session independently. The transformer configuration includes a hidden size of 64 and 4 attention heads. We use the Adam optimizer with a learning rate of 0.001 and eps set to 1e-7, without gradient clipping. For Amazon dataset, short-term sequences consist of the most recent 10 behaviors and long-term sequences the most recent 90. For Taobao dataset, short-term sequences include 16 behaviors and long-term sequences 256. The batch size is 512 on the industrial dataset, and 256 on both Taobao, Amazon and Ali Ads.

Figure 5 shows the online deployment of our generative interest fusion method, iFusion, in CTR prediction for a real-world industrial advertising system. iFusion constructs training data based on historical backflow logs to update the model. During online service, iFusion accepts inputs including user static features, ad features, user behavior sequence features, and candidate ad information. It outputs an estimated pCTR value, which is used for eCPM calculations in the downstream advertising system.

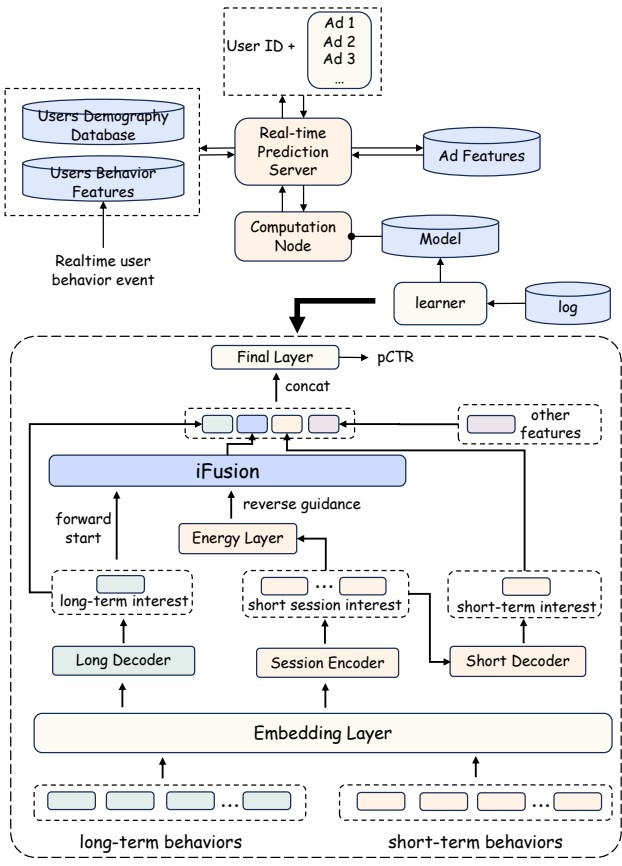

Figure 5: Real-time online deployment diagram of iFusion

## I ADDITIONAL HYPER-PARAMETER ANALYSIS FOR IFUSION

### I.1 THE IMPACT OF DIFFERENT GUIDANCE STRENGTH

We conducted hyperparameter experiments on the DCFG guidance strength hyperparameter $\gamma$, with candidate ranges for both $\gamma_{cp}$ and $\gamma_{tf}$ ranging from [0, 0.2, 0.4, 0.6, 0.8, 1.0, 2.0, 3.0, 4.0, 5.0], for a total of 25 combinations. Our results show that our method achieves the best results (AUC=0.7685) when $\gamma_{cp}$ is set to 2.0 and $\gamma_{tf}$ is set to 1.0, leading us to conduct additional comparison and ablation experiments based on this. We also found that when $\gamma_{cp}$ is set to 1, meaning that no core interest

guidance is included, our method performs the worst among all settings, and this performance increases with increasing $\gamma_{tf}$. Setting the $\gamma_{cp}$ to $\gamma_{tf}$ parameter ratio around 2.0 achieves relatively optimal results.

## J    EFFICIENCY ANALYSIS OF IFUSION

We analyze the efficiency of our proposed method iFusion from both offline training and online inference perspectives. Compared to our online base model, our diffusion-based approach incurs a modest 0.97% increase in training time on A100 GPU due to its iterative denoising process. For online serving, the step-wise generation introduces only a 0.302% latency increase while maintaining real-time responsiveness. These computational overheads are well justified by the significant improvements in recommendation quality, with significant AUC gain offline and CTR improvement online. The marginal efficiency costs are practically negligible in production environments, making our method highly feasible for real-world deployment. Training time and GPU memory utilization are shown in Table 4.

Table 4: More Efficiency Information

|  | base model | iFusion(ours) |
|---|---|---|
| Train Memory (GB) | 63.6 | 63.7 |
| Training GPU utilization | 79.50% | 79.63% |
| Inference Memory (GB) | 9.5 | 9.7 |
| Inference GPU utilization | 11.87% | 12.12% |

## K    PROOF OF THEOREM 1: ENERGY-BASED DISENTANGLEMENT

In this appendix, we provide a detailed proof of Theorem 1 from Section 4.1, which establishes the theoretical foundation for our Disentangled Classifier-Free Guidance (DCFG) framework.

### K.1    PRELIMINARIES

Recall that we model the conditional generation process using an energy-based formulation:

$$p(\mathbf{x}_t|g) \propto \exp(-E(\mathbf{x}_t|g)) = \exp(-[\gamma_{cp}E_{cp}(\mathbf{x}_t|g) + \gamma_{tf}E_{tf}(\mathbf{x}_t|g)]) \tag{43}$$

where $E_{cp}$ and $E_{tf}$ are energy functions implemented through architecturally constrained networks.

The score function is defined as the gradient of the log-probability:

$$\nabla_{\mathbf{x}_t} \log p(\mathbf{x}_t|g) = -\nabla_{\mathbf{x}_t} E(\mathbf{x}_t|g) \tag{44}$$

### K.2    PROOF OF SCORE DECOMPOSITION

From the definition of the total energy function:

$$E(\mathbf{x}_t|g) = \gamma_{cp}E_{cp}(\mathbf{x}_t|g) + \gamma_{tf}E_{tf}(\mathbf{x}_t|g) \tag{45}$$

$$\nabla_{\mathbf{x}_t} E(\mathbf{x}_t|g) = \gamma_{cp}\nabla_{\mathbf{x}_t} E_{cp}(\mathbf{x}_t|g) + \gamma_{tf}\nabla_{\mathbf{x}_t} E_{tf}(\mathbf{x}_t|g) \tag{46}$$

Substituting into the score function definition:

$$\begin{aligned} \nabla_{\mathbf{x}_t} \log p(\mathbf{x}_t|g) &= -\nabla_{\mathbf{x}_t} E(\mathbf{x}_t|g) \\ &= -\left(\gamma_{cp}\nabla_{\mathbf{x}_t} E_{cp}(\mathbf{x}_t|g) + \gamma_{tf}\nabla_{\mathbf{x}_t} E_{tf}(\mathbf{x}_t|g)\right) \\ &= \gamma_{cp}(-\nabla_{\mathbf{x}_t} E_{cp}(\mathbf{x}_t|g)) + \gamma_{tf}(-\nabla_{\mathbf{x}_t} E_{tf}(\mathbf{x}_t|g)) \end{aligned} \tag{47}$$

This completes the first part of the theorem, showing the exact decomposition of the conditional score function.

### K.3 Proof of Functional Disentanglement

We now prove that under appropriate architectural constraints, the guidance directions become functionally disentangled.

Let $\mathbf{v}_{cp} = -\nabla_{\mathbf{x}_t} E_{cp}$ and $\mathbf{v}_{tf} = -\nabla_{\mathbf{x}_t} E_{tf}$ represent the guidance directions for core preferences and transient fluctuations, respectively. The inner product between these directions is:

$$\langle \mathbf{v}_{cp}, \mathbf{v}_{tf} \rangle = \mathbf{v}_{cp}^\top \mathbf{v}_{tf} \tag{48}$$

We analyze the conditions under which this inner product is bounded by a small value $\zeta$. Consider the Taylor expansion of the energy functions around a point $\mathbf{x}_t^0$:

$$E_{cp}(\mathbf{x}_t) \approx E_{cp}(\mathbf{x}_t^0) + \nabla E_{cp}(\mathbf{x}_t^0)^\top (\mathbf{x}_t - \mathbf{x}_t^0) + \frac{1}{2}(\mathbf{x}_t - \mathbf{x}_t^0)^\top \mathbf{H}_{cp}(\mathbf{x}_t - \mathbf{x}_t^0) \tag{49}$$

$$E_{tf}(\mathbf{x}_t) \approx E_{tf}(\mathbf{x}_t^0) + \nabla E_{tf}(\mathbf{x}_t^0)^\top (\mathbf{x}_t - \mathbf{x}_t^0) + \frac{1}{2}(\mathbf{x}_t - \mathbf{x}_t^0)^\top \mathbf{H}_{tf}(\mathbf{x}_t - \mathbf{x}_t^0) \tag{50}$$

where $\mathbf{H}_{cp}$ and $\mathbf{H}_{tf}$ are the Hessian matrices of $E_{cp}$ and $E_{tf}$, respectively. The architectural constraints imposed on our networks ensure that:

1. $E_{cp}$ has low-pass characteristics, meaning $\mathbf{H}_{cp}$ has dominant eigenvalues corresponding to slow-varying directions in the input space.

2. $E_{tf}$ has high-pass characteristics, meaning $\mathbf{H}_{tf}$ has dominant eigenvalues corresponding to high-frequency directions in the input space.

Let $\{\mathbf{u}_1^{cp}, \mathbf{u}_2^{cp}, \ldots, \mathbf{u}_n^{cp}\}$ and $\{\lambda_1^{cp}, \lambda_2^{cp}, \ldots, \lambda_n^{cp}\}$ be the eigenvectors and eigenvalues of $\mathbf{H}_{cp}$, respectively. Similarly, let $\{\mathbf{u}_1^{tf}, \mathbf{u}_2^{tf}, \ldots, \mathbf{u}_n^{tf}\}$ and $\{\lambda_1^{tf}, \lambda_2^{tf}, \ldots, \lambda_n^{tf}\}$ be the eigenvectors and eigenvalues of $\mathbf{H}_{tf}$, respectively.

The architectural constraints ensure that the dominant eigenspaces of $\mathbf{H}_{cp}$ and $\mathbf{H}_{tf}$ are approximately orthogonal:

$$\langle \mathbf{u}_i^{cp}, \mathbf{u}_j^{tf} \rangle \approx 0 \quad \text{for } i \leq k, j \leq l \tag{51}$$

where $k$ and $l$ are the indices of the dominant eigenvalues for each Hessian.

Now, consider the gradients:

$$\nabla E_{cp} \approx \mathbf{H}_{cp}(\mathbf{x}_t - \mathbf{x}_t^0) \tag{52}$$

$$\nabla E_{tf} \approx \mathbf{H}_{tf}(\mathbf{x}_t - \mathbf{x}_t^0) \tag{53}$$

The inner product of the guidance directions is:

$$\begin{aligned} \langle \mathbf{v}_{cp}, \mathbf{v}_{tf} \rangle &= \langle -\nabla E_{cp}, -\nabla E_{tf} \rangle \\ &\approx (\mathbf{x}_t - \mathbf{x}_t^0)^\top \mathbf{H}_{cp}^\top \mathbf{H}_{tf}(\mathbf{x}_t - \mathbf{x}_t^0) \end{aligned} \tag{54}$$

Expanding $(\mathbf{x}_t - \mathbf{x}_t^0)$ in the eigenbases:

$$\mathbf{x}_t - \mathbf{x}_t^0 = \sum_{i=1}^n a_i \mathbf{u}_i^{cp} = \sum_{j=1}^n b_j \mathbf{u}_j^{tf} \tag{55}$$

Then:

$$\begin{aligned} \langle \mathbf{v}_{cp}, \mathbf{v}_{tf} \rangle &\approx \left( \sum_{i=1}^n a_i \mathbf{u}_i^{cp} \right)^\top \mathbf{H}_{cp}^\top \mathbf{H}_{tf} \left( \sum_{j=1}^n b_j \mathbf{u}_j^{tf} \right) \\ &= \sum_{i=1}^n \sum_{j=1}^n a_i b_j (\mathbf{u}_i^{cp})^\top \mathbf{H}_{cp}^\top \mathbf{H}_{tf} \mathbf{u}_j^{tf} \end{aligned} \tag{56}$$

Since $\mathbf{H}_{cp}$ and $\mathbf{H}_{tf}$ are symmetric (for twice-differentiable energy functions), we have:

$$(\mathbf{u}_i^{cp})^\top \mathbf{H}_{cp}^\top = \lambda_i^{cp} (\mathbf{u}_i^{cp})^\top \tag{57}$$

$$\mathbf{H}_{tf} \mathbf{u}_j^{tf} = \lambda_j^{tf} \mathbf{u}_j^{tf} \tag{58}$$

Thus:

$$\langle \mathbf{v}_{cp}, \mathbf{v}_{tf} \rangle \approx \sum_{i=1}^{n} \sum_{j=1}^{n} a_i b_j \lambda_i^{cp} \lambda_j^{tf} (\mathbf{u}_i^{cp})^\top \mathbf{u}_j^{tf} \tag{59}$$

Due to the approximate orthogonality of the dominant eigenspaces, the terms $(\mathbf{u}_i^{cp})^\top \mathbf{u}_j^{tf}$ are small for the indices $i, j$ where $\lambda_i^{cp}$ and $\lambda_j^{tf}$ are large. The remaining terms involve at least one small eigenvalue, making the product $\lambda_i^{cp} \lambda_j^{tf}$ small.

Therefore, there exists a bound $\zeta$ such that:

$$\langle \mathbf{v}_{cp}, \mathbf{v}_{tf} \rangle \leq \zeta \tag{60}$$

This proof demonstrates that through appropriate architectural constraints that shape the Hessian matrices of the energy functions, we can ensure that the guidance directions become approximately orthogonal. This functional disentanglement occurs without requiring strict conditional independence between the behavioral components, making our approach more robust and practical for real-world CTR prediction tasks. The value of $\zeta$ depends on the degree of orthogonality between the dominant eigenspaces of $\mathbf{H}_{cp}$ and $\mathbf{H}_{tf}$, which in turn is controlled by the architectural choices (e.g., pooling vs. attention) and regularization strategies employed in our framework.

## L    MORE DETAILS OF DCFG IMPLEMENTATION

Table 5: Key components of the DCFG framework implementation.

| Component | Implementation |
| --- | --- |
| Core Preference Encoder | Average-Pooling + MLP (strong regularization) |
| Transient Fluctuation Encoder | Attention + MLP (weak regularization) |
| Orthogonality Constraint | Gradient cosine similarity minimization |

## M    PROOFS OF ZERO-DATA THEOREM

### M.1    PROOF OF THEOREM 3

We consider the diffusion process defined by the forward SDE $dz = f(z,t)dt + g(t)dw$ and the reverse SDE $dz = [f(z,t) - g(t)^2 \nabla_z \log p_t(z)]dt + g(t)d\bar{w}$. The score function $\nabla_z \log p_t(z)$ captures the local geometry of the data distribution. On the manifold $\mathcal{M}$, the score function points toward regions of high data density, effectively guiding the reverse process along the manifold. By the manifold hypothesis, the data distribution $p_{\text{data}}$ is concentrated on $\mathcal{M}$, so the learned score function satisfies:

$$\nabla_z \log p_t(z) \approx \text{Proj}_{T_z \mathcal{M}}(\nabla_z \log p_t(z)) \tag{61}$$

where $\text{Proj}_{T_z \mathcal{M}}$ is the projection onto the tangent space.

The denoising process aims to recover $z_0$ from noisy observation $z_t$. The expected denoising error can be bounded:

$$\begin{aligned} \mathbb{E}[\|z_0 - \hat{z}_0\|^2] &\leq C_1 \cdot \mathbb{E}[\|\nabla_z \log p_t(z) - f_\theta(z,t)\|^2] \\ &\leq C_2 \cdot \mathbb{E}[\|\epsilon - \epsilon_\theta(z_t,t)\|^2] + \epsilon_{\text{approx}} \end{aligned} \tag{62}$$

where $f_\theta$ is the learned score function. Since the true data lies on $\mathcal{M}$, and the denoising process is consistent:

$$d_{\mathcal{M}}(z_0, z^*) \leq L \cdot \|z_0 - z^*\| \leq L \cdot (\|z_0 - \hat{z}_0\| + \|\hat{z}_0 - z^*\|) \tag{63}$$

In zero-data scenarios, the conditional information $z_l, z_s$ is uninformative. The diffusion process falls back to the unconditional score function $\nabla_z \log p_t(z)$, which still guides the generation toward the interest manifold $\mathcal{M}$.

## M.2 PROOF OF THEOREM 4

In the unconditional setting (zero behavioral data), the optimal denoising function minimizes:

$$\mathcal{L}(\theta) = \mathbb{E}_{z_0,\epsilon,t}[\|\epsilon - \epsilon_\theta(z_t, t)\|^2] \tag{64}$$

The solution to this optimization problem is given by:

$$\epsilon_\theta^*(z_t, t) = \mathbb{E}[\epsilon|z_t] = \mathbb{E}_{z_0 \sim p_{\text{data}}}[\epsilon|z_t] \tag{65}$$

This follows from Tweedie's formula and the Gaussian perturbation structure of the forward process:

$$z_t = \sqrt{\bar{\alpha}_t}z_0 + \sqrt{1 - \bar{\alpha}_t}\epsilon \tag{66}$$

Thus, in zero-data conditions, the denoising network learns to predict the expected noise given the noisy input, effectively performing Bayesian estimation using population-level statistics. The conditional case with uninformative $z_l, z_s$ reduces to the unconditional setting.

## N    EXPERIMENT ON THE EFFECT OF IFUSION IN ZERO-DATA SCENARIOS

We evaluate iFusion's robustness under two dimensions of user behavior sparsity: (1) absence of historical long-term behavior, and (2) scarcity of recent interaction records.

For the first scenario, we simulate missing historical data by deliberately omitting long-term interest representations during evaluation. Comparative analysis shows that iFusion's generative interest fusion module effectively compensates for the lack of long-term context, achieving an AUC gain of approximately 0.003 compared to traditional discriminative methods. This demonstrates the model's ability to infer stable interest representations despite incomplete historical data.

For the second scenario, we systematically reduce the number of available sessions and pad the corresponding interest representations. Traditional discriminative fusion methods exhibit significant performance drops due to their reliance on complete recent interaction sequences. In contrast, iFusion's generative paradigm substantially mitigates this sensitivity through its sampling-based generation process, which derives plausible interest points by leveraging the learned fused interest distribution and available long-term interest patterns. This approach maintains robust performance even with limited recent behavioral data.

## O    FEATURE MISALIGNMENT IN LONG-TERM AND SHORT-TEM INTEREST MODELING

The challenge of feature misalignment in long and short-term interest modeling is acknowledged in related work (Shen et al., 2022; Zheng et al., 2022). Prior work (Shen et al., 2022; Zheng et al., 2022) explicitly discusses the challenges of integrating heterogeneous user behaviors, where different interaction types like clicks and purchases carry distinct semantic meanings and naturally create feature space discrepancies between long-term and short-term sequences. Additionally, this misalignment stems from fundamental constraints in industrial deployment. In practice, long-term sequences typically contain only click behaviors due to latency constraints in online serving, while short-term sequences incorporate diverse behaviors (clicks, purchases, cart additions) to capture recent dynamic interests. This engineering reality creates an inherent heterogeneity between long-term and short-term feature spaces. Furthermore, users exhibit varying dependencies on long-term versus short-term interests across different contexts, making effective fusion even more challenging.

## P    THEOREM 1'S ORTHOGONALITY

The orthogonality of core signals and transient fluctuations within a session primarily impacts DCFG module without propagating system-wide constraints. Several works, such as End4rec (Han et al., 2024), SdifRec (Xie et al., 2024) and FMLP-Rec (Zhou et al., 2022), have proposed similar assumptions regarding the orthogonality of core signals and transient fluctuations. The disentanglement loss

$\mathcal{L}_{\text{Dist}}$ ( equation 18) explicitly minimizes the inner product between the core and fluctuation guidance vectors during training, which is a direct implementation of the orthogonality constraint. Its positive contribution in our ablation study implicitly validates this design. And we conduct an additional experiment to directly measure the cosine similarity between the guidance gradients $-\nabla E_{cp}$ and $-\nabla E_{tf}$. The results on a held-out validation set show a consistently low average cosine similarity of less than 0.102, providing concrete empirical evidence for the "approximate orthogonality" asserted in Theorem 1. We monitor this similarity throughout the training process. The value rapidly decreased and stabilized around the low level after a few batch, demonstrating that the disentanglement is not transient but a stable outcome of our training objective and architectural design.

## Q    SESSION MUTUAL INFORMATION IN THEOREM 2

We estimate the session mutual information $I(s_i; s_j)$ on our datasets. Given that short-term sessions are segmented from recent continuous behaviors, they naturally reflect similar contextual factors (e.g., promotional influences, purchasing power), leading to positive mutual information. However, due to the diverse intents and preference fluctuations across sessions, the mutual information values are estimated to be at a moderate to low level. This precisely aligns with the dependency condition in Theorem 2 ($I(s_i; s_j) > 0$), where AR injection provides significant benefits over NAR methods, while the moderate-low value explains why the advantage grows superlinearly rather than linearly.

## R    EXPERIMENTS ON TRAINING EPOCHS

Single epoch is the mainstream setup in research related to CTR prediction (Zhang et al., 2022; Cao et al., 2022; Liu et al., 2023b; Yan et al., 2024). As documented in previous studies (Zhang et al., 2022; Cao et al., 2022; Liu et al., 2023b; Yan et al., 2024), CTR models often exhibit one-epoch phenomenon where models begin to overfit after the first epoch, with test set performance degrading sharply upon entering subsequent training cycles. To rigorously validate this pattern in our experimental setup, we conduct comprehensive multi-epoch verification across all compared methods. The results consistently show that while all models achieved their performance peak within the first epoch, each method exhibit varying degrees of performance degradation when training continued through additional epochs. This empirical evidence confirms that single-epoch training not only aligns with CTR prediction task standards but also ensures we compare all methods at their optimal performance point. We have included the complete multi-epoch analysis in the appendix in our revised version.

Table 6: Performance comparison across different epochs

| Method | Epoch 1 | Epoch 2 | Epoch 3 |
|---|---|---|---|
| AvgPooling | 0.7512 | 0.7491 | 0.7457 |
| DIN | 0.7564 | 0.7548 | 0.7507 |
| DIEN | 0.7611 | 0.7589 | 0.7550 |
| SIM | 0.7625 | 0.7601 | 0.7574 |
| ETA | 0.7625 | 0.7603 | 0.7575 |
| SDIM | 0.7628 | 0.7604 | 0.7576 |
| TWIN | 0.7630 | 0.7607 | 0.7580 |
| TWIN-V2 | 0.7634 | 0.7609 | 0.7581 |
| MTGR | 0.7648 | 0.7619 | 0.7687 |
| DiffuRec | 0.7607 | 0.7582 | 0.7544 |
| DreamRec | 0.7619 | 0.7595 | 0.7558 |
| DiffuMIN | 0.7623 | 0.7596 | 0.7560 |
| iFusion | 0.7685 | 0.7652 | 0.7618 |

## S    Sensitivity to Diffusion Steps T

The training $T$ is set to 100 for all experiments. Due to the consistency distillation, the performance of our one-step inference model is remarkably robust to the original choice of $T$. As shown in Table 7, varying training $T$ has minimal impact on the final one-step inference performance.

Table 7: Performance comparison across different training steps $T$ and datasets

| Training $T$ | AUC (Industrial) | AUC (Amazon) | AUC (Taobao) | AUC (Ali Ads) |
|---|---|---|---|---|
| 50 | 0.7683 | 0.8508 | 0.9344 | 0.6651 |
| 100 | 0.7685 | 0.8512 | 0.9347 | 0.6652 |
| 200 | 0.7684 | 0.8510 | 0.9343 | 0.6649 |

## T    Generalizability Across Model Architectures

Using a simple DNN tower as the final prediction layer is a standard and controlled practice in this line of research (e.g., in DIN, DIEN, SIM) because it prevents the effects of a complex prediction tower from confounding the evaluation of the core behavior modeling module. iFusion framework is modular and agnostic to the final prediction tower. Its output, the fused interest representation $h_{fusion}$, is a unified vector that can serve as input to any downstream CTR model. To demonstrate broader applicability of iFusion, we conduct experiments integrating the iFusion-generated representations with DeepFM, DCN and AutoInt backbones. The results consistently show that iFusion provides a significant and stable performance lift over the baseline behavior fusion methods when used with these advanced architectures.

Table 8: Generalizability of iFusion Across Model Architectures

| Backbone | Method | AUC | RelImpr |
|---|---|---|---|
| DeepFM | MTGR | 0.7652 | 0.00% |
| DeepFM | iFusion | 0.7690 | +1.43% |
| DCN | MTGR | 0.7658 | 0.00% |
| DCN | iFusion | 0.7697 | +1.47% |
| AutoInt | MTGR | 0.7663 | 0.00% |
| AutoInt | iFusion | 0.7704 | +1.54% |

Table 9: Experiment of LTI and SSI Encoder

| LLI/SSI | iFusion | AUC |
|---|---|---|
| Att-based | w/ | 0.7685 |
| Att-based | w/o | 0.7641 |
| w/ AvgPooling | w/ | 0.7639 |
| w/ AvgPooling | w/o | 0.7602 |

## U    Details about LTI and SSI

In our implementation, both the long-term and short-term encoders are based on the widely adopted attention-based architectures commonly used for user behavior sequence modeling in CTR prediction. This choice ensures a fair and practical comparison with existing state-of-the-art methods. The iFusion framework, is designed to be agnostic to the underlying encoder architectures. The primary role of these encoders is to provide a competent initial representation of user interests. We conduct additional ablation studies where we varied the base architectures of the LTI and SSI encoders. The results consistently show that while the absolute performance baseline shifts with different encoders, the iFusion module consistently provides a stable and significant performance boost across these

variations. This demonstrates that the performance gains are primarily attributable to our novel fusion paradigm rather than to a specific encoder design. Therefore, while the encoders are necessary components to process raw behavior sequences, the specific choice of their architecture is not the critical factor for iFusion's superior performance.

