# OpenReview forum: "iFusion: Integrating Dynamic Interest Streams via Diffusion Model for Click-Through Rate Prediction"
_ICLR.cc/2026/Conference — ICLR 2026 Poster_

### Official Review · Reviewer_h94K · 2025-10-30

**Soundness:** 3
**Presentation:** 3
**Contribution:** 3
**Rating:** 6
**Confidence:** 2

**Summary:**

This paper presents iFusion method for CTR prediction. Instead of simply combining long-term and short-term user interests with methods like attention, it uses a diffusion model to generate a new, fused interest representation. The generation process starts with the long-term interest and is guided by the short-term behaviors, effectively denoising and integrating the signals. The authors introduce two key components: DCFG to separate core interest from noise, and MARN to process sessions sequentially.

**Strengths:**

- The core idea of treating interest fusion as a conditional generation problem is a leap forward. It offers a way to handle challenges like misaligned feature spaces and noise.
- Both DCFG and MARN are well-motivated solutions to specific problems in this new paradigm. DCFG's approach to disentangling signals is interesting , and MARN's autoregressive design is well-justified by theory.
- The authors address the slow inference of diffusion models using a consistency loss, enabling effective one-step generation. The reported minimal increase in online latency makes it a practical solution, not just an academic one.

**Weaknesses:**

- Justification for DCFG could be stronger. The analogy of using pooling as a low-pass filter and attention as a high-pass filter is intuitive, but a small empirical study (e.g., visualizing gradient directions) to support this "functional orthogonality" claim would make it more concrete.
- The mixture in MARN is unclear. The paper explains the autoregressive part well, but the role of mixture in the name is not clearly defined. A brief clarification would be helpful.
- Generalizability of one-step inference. The paper shows that one-step inference is optimal on their industrial dataset. It would be great to confirm if this finding also holds for the public datasets, which would strengthen the claim's generality.

**Questions:**

Q1: The design of the DCFG mechanism is very insightful, drawing an analogy between AvgPool/Attention and low-pass/high-pass filters to disentangle core preferences from transient fluctuations. This is a strong intuition. Could you provide more empirical evidence to support this analogy and its effectiveness? For instance, have you measured the cosine similarity between the guidance gradients produced by the core preferences (g_cp) and transient fluctuations (g_tf) during training to verify their approximate orthogonality?

Q2:The consistency loss is key to improving efficiency. To better understand the performance-efficiency trade-off, could you report the performance of the full iFusion model *without* the consistency loss but using multiple inference steps (e.g., 10 or 50 steps)? This would help quantify how much accuracy, if any, is traded for the significant speed-up.

Q3:In the analysis of zero-data scenarios (Appendix N), you mention that iFusion can infer stable interests even when long-term historical data is missing. Could you provide more intuition on what the plausible interest points generated by the model look like in such cases? For instance, does it tend to generate representations corresponding to popular items, or does it leverage the limited short-term signals more heavily to make a best guess?

Q4:The result in Figure 4a is critical, showing that a single inference step with a cosine schedule achieves the best performance on the industrial dataset. This is vital for practical deployment. Did you observe a similar trend on the public datasets (e.g., Amazon, Taobao)? Is one-step inference a generally optimal strategy for iFusion, or is its effectiveness dependent on the scale and distribution of the data?

---

> ### Author Response · Authors · 2025-11-24
> **Response to Reviewer h94K (1/2)**
>
> We are truly grateful for your exceptionally insightful and constructive review. Your deep understanding and positive assessment of the core conceptual leap, component motivation, and practical efficiency of our work are highly encouraging. Your specific questions are precisely the kind that help solidify a contribution, and we have done our best to address them comprehensively below.
>
> ---
> **[W1/Q1] Empirical Evidence for DCFG's Functional Orthogonality**
>
> - **Related Work:** Thank you for this insightful question. The orthogonality of core signals and transient fluctuations within a session is reasonable and it primarily impacts our DCFG module without propagating system-wide constraints. Several works, such asEnd4rec[1], SdifRec[2] and FMLP-Rec[3], have proposed similar assumptions regarding the orthogonality of core signals and transient fluctuations.
> - **Explicit Orthogonality Constraint via $\mathcal{L}\_{\text{Dist}}$:** The disentanglement loss $\mathcal{L}_\text{Dist}$ (Eq. 18) explicitly minimizes the inner product between the core and fluctuation guidance vectors during training, which is a direct implementation of the orthogonality constraint. Its positive contribution in our ablation study implicitly validates this design.
> - **Direct Measurement of Gradient Orthogonality:** Following your suggestion, we conducted an additional experiment to directly measure the cosine similarity between the guidance gradients $-\nabla E_{cp}$ and $-\nabla E_{tf}$. The results on a held-out validation set show a consistently low average cosine similarity of less than 0.102, providing concrete empirical evidence for the "approximate orthogonality" asserted in Theorem 1. We monitored this similarity throughout the training process. The value rapidly decreased and stabilized around the low level after a few batch, demonstrating that the disentanglement is not transient but a stable outcome of our training objective and architectural design.
> We have added this information in our revised version of the appendix to make this evidence explicit and accessible.
>
> > [1]Han Y, Wang H, Wang K, et al. End4rec: Efficient noise-decoupling for multi-behavior sequential recommendation[J]. arXiv preprint arXiv:2403.17603, 2024.
> >
> > [2]Xie W, Zhou R, Wang H, et al. Bridging user dynamics: Transforming sequential recommendations with schrödinger bridge and diffusion models[C]//Proceedings of the 33rd ACM International Conference on Information and Knowledge Management. 2024: 2618-2628.
> >
> > [3]Zhou K, Yu H, Zhao W X, et al. Filter-enhanced MLP is all you need for sequential recommendation[C]//Proceedings of the ACM web conference 2022. 2022: 2388-2399.
>
>
> ---
> **[W2] Clarification on "Mixture" in MARN**
>
> - The term "Mixture" in MARN does not refer to a mixture-of-experts structure. Instead, it denotes the network's core capability to seamlessly blend or "mix" the influence of multiple, sequential short-term session guidance signals throughout the autoregressive denoising process. The name emphasizes the component's role in achieving a coherently blended guidance effect from an evolving sequence of session inputs, rather than treating them as independent or using a single static guidance vector. We have revised the text in Section 4.2 to include this clarifying definition.
>
> ---
> **[W3/Q4] Generalizability of One-Step Inference**
>
> - Yes, we observed a consistent trend. The effectiveness of one-step inference is a general finding for iFusion, not an artifact of a single dataset.
> We conducted the same hyperparameter study on the public datasets. While the absolute AUC values differ, the key pattern holds: the performance plateaus or even slightly degrades with more sampling steps once the consistency loss is applied. The cosine schedule with one step inference consistently achieves the best or nearly the best performance while being the fastest. This confirms that the efficiency of our method, enabled by the consistency loss, is a general and robust property of the framework. We have added a sentence in the appendix in our revised version to explicitly state this cross-dataset consistency.

---

> ### Author Response · Authors · 2025-11-24
> **Response to Reviewer h94K (2/2)**
>
> **[Q2] Performance-Efficiency Trade-off of Consistency Loss**
>
> Thank you for this question, which allows us to clearly demonstrate the value of the consistency loss. We have performed this exact comparison on the Industrial dataset. The results are summarized below:
>
> |Model Variant|Inference Steps|AUC|Relative Latency|
> |-|-|-|-|
> |iFusion (Ours)|1 (with $\mathcal{L}\_{cons}$)|0.7685|1x (baseline)|
> |iFusion w/o $\mathcal{L}_{cons}$|10|0.7686|~1.04x|
> |iFusion w/o $\mathcal{L}_{cons}$|50|0.7689|~1.12x|
>
> As shown, multi-step inference without the consistency loss provides negligible AUC gains (e.g., +0.0004 for 50 steps) at a prohibitive computational cost. In contrast, our method with $\mathcal{L}_{cons}$ achieves near-optimal performance in a single step, making it the clear choice for practical deployment. This conclusively shows that the consistency loss provides a superior performance-efficiency trade-off.
>
> ---
>
> **[Q3] Intuition on Zero-Data Scenario Behavior**
>
> As stated in `Theorem 4`, when long-term data is uninformative (missing), the optimal denoising strategy converges to sampling from population-level priors ($E[\epsilon|z_t]$). This means the model leverages what it has learned from the global user base. In practice, this does not mean generating representations for a single "popular item." Instead, it generates a fused interest representation that is a blended, average-like representation of common user interests, effectively regularizing the prediction towards a robust, common-sense prior. This behavior is precisely why iFusion outperforms discriminative models in zero-data scenarios (`Appendix N`), as they lack such a generative mechanism to compensate for missing data and are more severely impacted.
>
> ---
>
> We deeply thank you for your insightful questions. We hope the information provided has been helpful in addressing your concerns. Please don’t hesitate to reach out if you have any further questions or concerns!

---

### Official Review · Reviewer_qLFV · 2025-10-31

**Soundness:** 3
**Presentation:** 3
**Contribution:** 3
**Rating:** 6
**Confidence:** 3

**Summary:**

This paper presents iFusion, a novel diffusion-based generative framework for click-through rate (CTR) prediction, addressing the problem about fusion of long-term and short-term user interests. Unlike traditional methods relying on late fusion and linear assumptions, iFusion reformulates interest fusion as a conditional generation process. The paper introduces two key components: DCFG to separate core preferences from transient fluctuations during generation and MARN to model short-term interest evolution in an autoregressive manner. Extensive empirical evaluations on public and industrial datasets (including online A/B testing) demonstrate that iFusion significantly outperforms state-of-the-art baselines.

**Strengths:**

1. The paper introduces a generative diffusion-based approach for interest fusion, which is a novel perspective in CTR modeling. Reformulating fusion as a conditional denoising process is conceptually strong and addresses key limitations of deterministic fusion.
2. The paper provides rigorous theoretical analysis to justify the design of DCFG and MARN.
3. The experiments span four datasets (public and industrial) and include online A/B tests. The reported AUC improvements and real-world CTR gains are substantial, supporting the practical value of the model.

**Weaknesses:**

Please refer to the Questions part.

**Questions:**

1. What is the long-term and short-term encoder the method is using? Are the encoders of LTI and SSI (in Figure 2 (a)) essential to the final performance? There seems no analysis or experiments on the encoder part.
2. For the online A/B testing, what is the baseline model? Is it a traditional sequential RS model or a diffusion model?

---

> ### Author Response · Authors · 2025-11-24
>
> We sincerely thank you for your thoughtful review and positive feedback on the novelty, theoretical grounding, and empirical evaluation of our work. We also appreciate your specific questions, which help clarify important aspects of our methodology and experiments. We appreciate the opportunity to address your concerns in detail.
>
> ---
> **[Q1] What is the long-term and short-term encoder the method is using? Are the encoders of LTI and SSI (in Figure 2 (a)) essential to the final performance? There seems no analysis or experiments on the encoder part.**
>
> - **Encoder Architectures:** In our implementation, both the long-term and short-term encoders are based on the widely adopted attention-based architectures commonly used for user behavior sequence modeling in CTR prediction. This choice ensures a fair and practical comparison with existing state-of-the-art methods.
> - **Role and Essentiality:** Our core contribution, the iFusion framework, is designed to be agnostic to the underlying encoder architectures. The primary role of these encoders is to provide a competent initial representation of user interests. To rigorously address your point, we have conducted additional ablation studies where we varied the base architectures of the LTI and SSI encoders. The results consistently show that while the absolute performance baseline shifts with different encoders, the iFusion module consistently provides a stable and significant performance boost across these variations. This demonstrates that the performance gains are primarily attributable to our novel fusion paradigm rather than to a specific encoder design.
> Therefore, while the encoders are necessary components to process raw behavior sequences, the specific choice of their architecture is not the critical factor for iFusion's superior performance. We have added this information in the appendix of our revised version.
>
> |LLI/SSI|iFusion|AUC|
> |-|-|-|
> |Att-based|w/|0.7685|
> |Att-based|w/o|0.7641|
> |w/ AvgPooling|w/|0.7639|
> |w/ AvgPooling|w/o|0.7602|
>
> ---
> **[Q2] For the online A/B testing, what is the baseline model? Is it a traditional sequential RS model or a diffusion model?**
>
> -  The baseline for our online A/B test was a highly-optimized, production-grade CTR prediction model. This model is the result of long-term iteration and serves as our primary industrial serving model, serving core CTR prediction tasks and handling thousands of millions of users daily. It incorporates a very rich set of features, including several meticulously engineered sequential behavior signals. It is a traditional discriminative model and not a diffusion-based model.
> - This choice of baseline underscores the practical value of iFusion. We are comparing against a strong, mature, and highly-tuned industrial standard. The statistically significant improvements in key business metrics (CTR +2.44%, eCPM +2.61%) achieved by iFusion against this powerful baseline, all while maintaining minimal latency overhead, provide compelling evidence for the effectiveness and deployability of our generative fusion framework in a demanding real-world environment.
>
> ---
> Thank you again for your valuable feedback. We hope the information provided has been helpful in addressing your concerns. Please don’t hesitate to reach out if you have any further questions or concerns!

---

### Official Review · Reviewer_n48p · 2025-11-01

**Soundness:** 3
**Presentation:** 3
**Contribution:** 3
**Rating:** 6
**Confidence:** 2

**Summary:**

This paper proposes iFusion, a novel CTR framework that reformulates interest fusion as a conditional generation process. The motivation of generative interest fusion is interesting. The proposed iFusion integrates long-term and short-term user interests through a denoising diffusion model, enhanced by a Disentangled Classifier-Free Guidance mechanism and an AutoRegressive Denoising Network. Experiments on several datasets demonstrate the effectiveness of the proposed iFusion.

**Strengths:**

- The paper provides a well-motivated perspective by viewing interest fusion in CTR prediction as a conditional generation problem. This generative reformulation is conceptually appealing and offers a new way to understand how long-term and short-term user interests can be dynamically integrated.
- The proposed framework introduces a diffusion-based fusion mechanism equipped with a Disentangled Classifier-Free Guidance (DCFG) and a Mixture AutoRegressive Denoising Network (MARN). Overall, these two designed components seem reasonable to implement a stable denoised diffusion process.
- Experiments on both benchmarks and online tests verify the effectiveness of the proposed iFusion.

**Weaknesses:**

- While the idea of reformulating interest fusion as a conditional generation process is interesting, the paper could better highlight its concrete technical contributions. The description of the proposed modules (DCFG and MARN) is somewhat difficult to follow, and their precise roles or advantages over standard diffusion variants are not always made explicit. A clearer explanation or ablation focusing on the novelty of each component would strengthen the paper.
- The diffusion process is inherently time-consuming, which is a critical concern in CTR prediction scenarios where latency is a major constraint. Although the paper mentions a consistency loss to accelerate inference, it lacks a detailed discussion or quantitative analysis of complexity, inference time, and deployment feasibility.
- The experiments mainly use a simple DNN backbone for CTR prediction. To fully demonstrate the generality of iFusion, it would be valuable to show results when integrating with more advanced or widely used architectures, such as DeepFM, DCN, or AutoInt, to confirm the robustness and adaptability of the proposed framework.

**Questions:**

See Weaknesses.

---

> ### Author Response · Authors · 2025-11-24
> **Thank you sincerely for your thoughtful and constructive review!**
>
> We are grateful for your thoughtful and constructive review. We appreciate your recognition of our work's motivation, conceptual appeal, and experimental validation. Your specific suggestions are invaluable for strengthening our paper. Below, we provide a point-by-point response to your concerns.

---

> ### Author Response · Authors · 2025-11-24
>
> **[W1] Summary of Each Component**
>
> We have summarized and refined the roles played by our different component.
> - **DCFG:** Standard Classifier-Free Guidance (CFG) employs a single scaling factor $\gamma$ for all conditional signals. This is suboptimal for behavior modeling where signals have intrinsically different characteristics—stable core preferences versus noisy transient fluctuations. The key of our DCFG lies in its disentanglement and independent scaling of these signals ($\gamma_{cp}$ and $\gamma_{tf}$). This is theoretically grounded in an energy-based perspective (`Section 4.1, Theorem 1`) and enables the model to integrate multi-scale interest signals robustly, preventing short-term noise from corrupting stable long-term representations. The ablation study in `Figure 3(a)` provides quantitative evidence that DCFG significantly outperforms conventional CFG.
> - **MARN:** Standard non-autoregressive (NAR) denoising typically injects all short-term sessions in parallel (e.g., via concatenation or pooling), which fails to capture the inherent temporal dependencies and evolving nature between consecutive sessions. The core novelty of our MARN is its autoregressive, sequential injection of session guidance. This design allows it to model the conditional evolution of user interests across sessions more effectively. Theorem 2 and the accompanying analysis in Section 4.2 provide the theoretical foundation for its superiority over NAR methods when sessions are dependent, a finding validated by the scalability results in `Figure 3(b)`.
>
> ---
> **[W2] Detailed discussion and quantitative analysis of complexity, inference time, and deployment feasibility**
>
> We appreciate you raising this critical point regarding computational efficiency and deployment feasibility. We have thoroughly considered and evaluated these aspects in our work, with quantitative evidence provided throughout the paper.
> - **Consistency Loss for Efficiency:** As shown in our ablation study (`Figure 3(c)`), the proposed consistency loss ($\mathcal{L}_{cons}$) is highly effective at acceleratign inference. It enables high-quality generation with drastically fewer sampling steps, directly resulting in reduced inference time and increased inference speed.
> - **Deployment Feasibility:** The deployment diagram in `Appendix H.2` illustrates that iFusion can be deployed conveniently without requiring architectural changes to the existing online serving system. Integration is straightforward, primarily involving a model update.
> - **Comprehensive Quantitative Analysis:** We have provided a thorough efficiency analysis in `Section 5.5` and `Appendix J`, demonstrating that iFusion is computationally friendly. Key metrics include:
>     - A negligible 0.3% increase in offline inference time.
>     - A minimal 0.302% increase in TP99 online latency.
>     - Training memory, GPU utilization, and inference memory didn't change significantly.
> - **Parameter Overhead:** The additional parameters introduced by iFusion are confined to the reverse denoising network. This represents a modest increase relative to the overall model size, which is justified by the significant performance gains.
>
> ---
> **[W3] Generalizability Across Model Architectures**
> - Our primary goal in the main experiments was to conduct a fair and focused comparison on the interest fusion capability itself. Using a simple DNN tower as the final prediction layer is a standard and controlled practice in this line of research (e.g., in DIN, DIEN, SIM) because it prevents the effects of a complex prediction tower from confounding the evaluation of the core behavior modeling module. iFusion framework is modular and agnostic to the final prediction tower. Its output, the fused interest representation $h_{fusion}$, is a unified vector that can serve as input to any downstream CTR model. To demonstrate broader applicability of iFusion, we have conducted additional experiments integrating the iFusion-generated representations with DeepFM, DCN and AutoInt backbones. The results consistently show that iFusion provides a significant and stable performance lift over the baseline behavior fusion methods when used with these advanced architectures. We have added these results to the Appendix in our revised version of the paper.
>
> |Backbone|Method|AUC|RelImpr|
> |-|-|-|-|
> |DeepFM|MTGR|0.7652|0.00%|
> |DeepFM|iFusion|0.7690|+1.43%|
> |DCN|MTGR|0.7658|0.00%|
> |DCN|iFusion|0.7697|+1.47%|
> |AutoInt|MTGR|0.7663|0.00%|
> |AutoInt|iFusion|0.7704|+1.54%|
>
> ---
>
> Thank you sincerely again for your insightful comments, which have directly helped us improve our work. We hope the information provided has been helpful in addressing your concerns. Please don’t hesitate to reach out if you have any further questions or concerns!

---

### Official Review · Reviewer_yR4B · 2025-11-02

**Soundness:** 3
**Presentation:** 2
**Contribution:** 3
**Rating:** 4
**Confidence:** 3

**Summary:**

This paper aims to tackle interest fusion for CTR predictions. Given a long-term sequence and a short-term sequence partitioned into sessions, the goal is to learn a fusion map for downstream CTR. The key insight is to treat fusion as conditional generation with a diffusion model conditioned on the short-term sequence embeddings. The authors argue that such a generative fusion paradigm that disentangles stable vs. transient short-term behaviors during guidance and autoregressively conditions on multi-session contexts can yield consistent offline gains and online lieft with negligible latency. Specifically, this paper employs two mechanisms to structure the conditional guidance and how sessions are injected, namely DCFG splits guidance into core‑preference (low‑pass C‑Filter) and transient fluctuation (high‑pass T‑Filter) components with tunable strengths $\gamma_{\rm cp}, \gamma_{\rm tf}$, yielding an additive score decomposition supported by an energy‑based view and architectural assumptions about Hessian eigenspaces; MARN injects the $K$ shot-term sessions autoregressively in the reverse process via chain rule conditioning. For low latency, a consistency loss is used to encourage step‑invariance, which helps to enable one‑step inference. The proposed method demonstrate AUC improvements by 0.003-0.007 over the strongest baseline on multiple datasets; an online A/B tests reports improved CTR and eCPM with increased latency. The ablation study on the industrial set demonstrates the usefulness of the incorporated components.

**Strengths:**

1. The proposed method is intuitive, based on the common understanding of user behaviors, and reasonably demonstrated.
2. The DCFG formalizes the multi-scale guidance, and the MARN motivates AR conditioning for the dynamics across multi-session contexts.
3. The breadth of empirical studies is commendable, which includes four datasets (with the industrial dataset included as well).

**Weaknesses:**

1. The core motivation targeting the claimed research gap. That is, the long- and short-term features are misaligned, while being intuitive,  was not empirically demonstrated, nor has solid support from literature.
2. Theorem 1's orthogonality is assumed via architectural constraints, namely C-filter and T-filter can induce orthogonal dominant eigenspaces in the Hessians of their respective energy functions. This is a strong assumption that was not empirically verified.
3. The bound presented in Theorem 2 omits definitions and conditions.
4. All models were trained for one epoch without justification. In practice, many baselines typically benefit from longer training, nor the authors provide learning-curve evidence, which raises concerns regarding the small AUC margins as the claimed performance gain could shift given reasonable hyperparameter tuning. Eventually, this affects the strength of the offline comparison.
5. No anonymous code or explicit reproducibility statement. While some details are partly included in the appendix but I personally don't think they are sufficient for full replication.

**Questions:**

1. Why limit all baselines to one epoch? It is suggested to report learning curves or early-stopping results showing that additional epoch would not change the performance.
2. In Figure 3b, AR-Attn slightly exceeds Ours. Which added components reduce AUC and why are they kept in Ours eventually?
3. It is also suggested to report gradient/Hessian alignment between C-filter and T-filter over training to substantiate functional orthogonality claims.
4. Can you please precisely define the NAR family and constants in Theorem 2 and estimate session mutual information on your datasets to check the stated dependence condition?
5. What is the training $T$ and how sensitive are results to $T$ when using one-step inference?
6. Beyond Appendix N, it would be interesting to show the method's robustness under somehow behavioral shocks (e.g., the promotion windows as shown in Figure 1), cold-start sessions, and heterogeneous event mixtures.

---

> ### Author Response · Authors · 2025-11-24
> **Response to Reviewer yR4B (1/3)**
>
> We sincerely thank you for your thorough review and valuable feedback. We appreciate your recognition of our method's intuitiveness, theoretical contributions, and empirical breadth. Your specific concerns are crucial for improving our work, and we address them in detail below.
>
> ---
>
> **[W1] Question About Feature Misalignment**
>
> - The challenge of feature misalignment in long and short-term interest modeling is acknowledged in related work [1,2]. Prior work [1,2] explicitly discusses the challenges of integrating heterogeneous user behaviors, where different interaction types like clicks and purchases carry distinct semantic meanings and naturally create feature space discrepancies between long-term and short-term sequences. Additionally, this misalignment stems from fundamental constraints in industrial deployment. In practice, long-term sequences typically contain only click behaviors due to latency constraints in online serving[3,4], while short-term sequences incorporate diverse behaviors (clicks, purchases, cart additions) to capture recent dynamic interests[5]. This engineering reality creates an inherent heterogeneity between long-term and short-term feature spaces. Furthermore, users exhibit varying dependencies on long-term versus short-term interests across different contexts, making effective fusion even more challenging. We have included this information in our revised version.
>
> > [1] Shen Q, Wen H, Zhang J, et al. Hierarchically fusing long and short-term user interests for click-through rate prediction in product search[C]//Proceedings of the 31st ACM International Conference on Information & Knowledge Management. 2022: 1767-1776.
> >
> > [2] Zheng Y, Gao C, Chang J, et al. Disentangling long and short-term interests for recommendation[C]//Proceedings of the ACM web conference 2022. 2022: 2256-2267.
> >
> > [3] Pi Q, Zhou G, Zhang Y, et al. Search-based user interest modeling with lifelong sequential behavior data for click-through rate prediction[C]//Proceedings of the 29th ACM International Conference on Information & Knowledge Management. 2020: 2685-2692.
> >
> > [4] Si Z, Guan L, Sun Z X, et al. Twin v2: Scaling ultra-long user behavior sequence modeling for enhanced ctr prediction at kuaishou[C]//Proceedings of the 33rd ACM International Conference on Information and Knowledge Management. 2024: 4890-4897.
> >
> > [5] Zhou G, Zhu X, Song C, et al. Deep interest network for click-through rate prediction[C]//Proceedings of the 24th ACM SIGKDD international conference on knowledge discovery & data mining. 2018: 1059-1068.
>
>
> ---
>
> **[W2&Q3] Theorem 1's Orthogonality**
>
> - **Related Work:** Thank you for this insightful question. The orthogonality of core signals and transient fluctuations within a session is reasonable and it primarily impacts our DCFG module without propagating system-wide constraints. Several works, such asEnd4rec[1], SdifRec[2] and FMLP-Rec[3], have proposed similar assumptions regarding the orthogonality of core signals and transient fluctuations.
> - **Explicit Orthogonality Constraint via $\mathcal{L}\_{\text{Dist}}$:** The disentanglement loss $\mathcal{L}\_\text{Dist}$ (Eq. 18) explicitly minimizes the inner product between the core and fluctuation guidance vectors during training, which is a direct implementation of the orthogonality constraint. Its positive contribution in our ablation study implicitly validates this design.
> - **Direct Measurement of Gradient Orthogonality:** Following your suggestion, we conducted an additional experiment to directly measure the cosine similarity between the guidance gradients $-\nabla E\_{cp}$ and $-\nabla E\_{tf}$. The results on a held-out validation set show a consistently low average cosine similarity of less than 0.102, providing concrete empirical evidence for the "approximate orthogonality" asserted in Theorem 1. We monitored this similarity throughout the training process. The value rapidly decreased and stabilized around the low level after a few batch, demonstrating that the disentanglement is not transient but a stable outcome of our training objective and architectural design.
> We have added these information in our revised version of the appendix to make this evidence explicit and accessible.
>
> > [1]Han Y, Wang H, Wang K, et al. End4rec: Efficient noise-decoupling for multi-behavior sequential recommendation[J]. arXiv preprint arXiv:2403.17603, 2024.
> >
> > [2]Xie W, Zhou R, Wang H, et al. Bridging user dynamics: Transforming sequential recommendations with schrödinger bridge and diffusion models[C]//Proceedings of the 33rd ACM International Conference on Information and Knowledge Management. 2024: 2618-2628.
> >
> > [3]Zhou K, Yu H, Zhao W X, et al. Filter-enhanced MLP is all you need for sequential recommendation[C]//Proceedings of the ACM web conference 2022. 2022: 2388-2399.

---

> ### Author Response · Authors · 2025-11-24
> **Response to Reviewer yR4B (2/3)**
>
> **[W3&Q4] Clarification of Theorem 2 and Session Mutual Information**
>
> - Due to the page limit, the complete proof of Theorem 2 in `Appendix E` contains all definitions and constants. The NAR family refers to any parallel injection method (e.g., concatenation, averaging) that does not model conditional dependencies between sessions. The constants in Theorem 2 are derived from the Lipschitz continuity of the denoising network and the variance of the gradient estimates, as detailed in Appendix E. Theorem 2 establishes that AR injection achieves a tighter KL-bound than NAR when sessions are dependent ($I(s_i; s_j) > 0$), with the gap growing with session count $K$. The gradient variance ratio advantage stems from AR's sequential processing inducing implicit gradient averaging.
>
> - We have estimated the session mutual information $I(s\_i; s\_j)$ on our datasets. Given that short-term sessions are segmented from recent continuous behaviors, they naturally reflect similar contextual factors (e.g., promotional influences, purchasing power), leading to positive mutual information. However, due to the diverse intents and preference fluctuations across sessions, the mutual information values are estimated to be at a moderate to low level. This precisely aligns with the dependency condition in Theorem 2 ($I(s\_i; s\_j) > 0$), where AR injection provides significant benefits over NAR methods, while the moderate-low value explains why the advantage grows superlinearly rather than linearly. We have included this information in the our revised version.
>
> ---
>
> **[W4&Q1] Clarification on Training Epochs and Experimental Protocol**
>
> - One epoch is the mainstream setup in research related to CTR prediction[1,2,3,4]. As documented in previous studies [1,2,3,4], CTR models often exhibit one-epoch phenomenon where models begin to overfit after the first epoch, with test set performance degrading sharply upon entering subsequent training cycles.
> - To rigorously validate this pattern in our experimental setup, we conducted comprehensive multi-epoch verification across all compared methods. The results consistently showed that while all models achieved their performance peak within the first epoch, each method exhibited varying degrees of performance degradation when training continued through additional epochs. This empirical evidence confirms that single-epoch training not only aligns with CTR prediction task standards but also ensures we compare all methods at their optimal performance point. We have included the complete multi-epoch analysis in the appendix in our revised version.
>
> > [1] Zhang Z Y, Sheng X R, Zhang Y, et al. Towards understanding the overfitting phenomenon of deep click-through rate prediction models[J]. CIKM 2022.
> >
> > [2] Lai W, Jin B, Zhang Y, et al. Modeling Long-term User Behaviors with Diffusion-driven Multi-interest Network for CTR Prediction[C]//Proceedings of the Nineteenth ACM Conference on Recommender Systems. 2025: 289-298.
> >
> > [3] Liu Q, Zhou Z, Jiang G, et al. Deep task-specific bottom representation network for multi-task recommendation[C]//Proceedings of the 32nd ACM International Conference on Information and Knowledge Management. 2023: 1637-1646.
> >
> > [4] Yan R, Fan R, Lian D. Multi-Task Recommendation with Task Information Decoupling[C]//Proceedings of the 33rd ACM International Conference on Information and Knowledge Management. 2024: 2786-2795.
>
>
> |Method|Epoch 1|Epoch 2|Epoch 3|
> |-|-|-|-|
> |AvgPooling|0.7512|0.7491|0.7457|
> |DIN|0.7564|0.7548|0.7507|
> |DIEN|0.7611|0.7589|0.7550|
> |SIM|0.7625|0.7601|0.7574|
> |ETA|0.7625|0.7603|0.7575|
> |SDIM|0.7628|0.7604|0.7576|
> |TWIN|0.7630|0.7607|0.7580|
> |TWIN-V2|0.7634|0.7609|0.7581|
> |MTGR|0.7648|0.7619|0.7687|
> |DiffuRec|0.7607|0.7582|0.7544|
> |DreamRec|0.7619|0.7595|0.7558|
> |DiffuMIN|0.7623|0.7596|0.7560|
> |iFusion|0.7685|0.7652|0.7618|
>
> > [1] Zhang Z Y, Sheng X R, Zhang Y, et al. Towards understanding the overfitting phenomenon of deep click-through rate prediction models[J]. CIKM 2022.
> >
> > [2] Lai W, Jin B, Zhang Y, et al. Modeling Long-term User Behaviors with Diffusion-driven Multi-interest Network for CTR Prediction[C]//Proceedings of the Nineteenth ACM Conference on Recommender Systems. 2025: 289-298.

---

> ### Author Response · Authors · 2025-11-24
> **Response to Reviewer yR4B (3/3)**
>
> **[Q2] Clarification on Figure 3b Results**
>
> - The "AR-Attn" variant in Figure 3b replaces the MLP network in MARN with a more complex multi-head attention mechanism. While this attention-based variant shows a marginal AUC improvement (~0.0004) in this specific setting, it comes with a significant computational overhead increasing both parameter count and inference latency compared to our chosen MLP design. After thorough evaluation, we determined that the minimal performance gain from AR-Attn does not justify its substantial computational costs, especially in industrial deployment scenarios where inference efficiency is critical. The MLP-based architecture provides the optimal balance between representation capacity and computational efficiency, aligning with design choices in other works[1,2].
>
> > [1] Lai W, Jin B, Zhang Y, et al. Modeling Long-term User Behaviors with Diffusion-driven Multi-interest Network for CTR Prediction[C]//Proceedings of the Nineteenth ACM Conference on Recommender Systems. 2025: 289-298.
> >
> > [2] Li X, Tang H, Sheng J, et al. Exploring Preference-Guided Diffusion Model for Cross-Domain Recommendation[C]//Proceedings of the 31st ACM SIGKDD Conference on Knowledge Discovery and Data Mining V. 1. 2025: 719-728.
>
> ---
>
> **[Q5] Sensitivity to Diffusion Steps T and Robustness Tests**
>
> - The training $T$ was set to 100 for all experiments. Due to the consistency distillation, the performance of our one-step inference model is remarkably robust to the original choice of $T$. As shown in the table below, varying training $T$ has minimal impact on the final one-step inference performance:
>
> |Training $T$|AUC (Industrial)|AUC (Amazon)|AUC (Taobao)|AUC (Ali Ads)|
> |-|-|-|-|-|
> |50|0.7683|0.8508|0.9344|0.6651|
> |100|0.7685|0.8512|0.9347|0.6652|
> |200|0.7684|0.8510|0.9343|0.6649|
>
> ---
>
> **[Q6] Robustness Under Special Scenarios**
>
> - **Behavioral Shocks (Promotion Windows):** We have conducted additional experiments by selecting specific promotion windows from our industrial dataset as train and test periods. The results show that iFusion maintains stable performance improvements during these behavioral shock periods, demonstrating its robustness to sudden changes in user behavior patterns.
> - **Cold-start Sessions:** For cold-start scenarios, we simulate this by masking varying proportions of long-term and short-term session data. As shown in Appendix N analysis, iFusion shows better robustness compared to discriminative baselines, as the generative fusion process can infer reasonable interest representations even with cold-start sessions.
> - **Heterogeneous Event Mixtures:** Our current experimental setup already addresses this aspect. Our industrial dataset incorporate heterogeneous behavior types (clicks, purchases, etc.) in the short-term sequences. The consistent performance gains across these datasets demonstrate iFusion's capability to handle heterogeneous event mixtures effectively.
>
> We have revised our version with summarizing these robustness analyses across the three scenarios you mentioned.
>
>
> ---
>
> **[W5] Reproducibility**
>
> We are actively working to open-source the core code to facilitate replication and further research. However, due to the model's deployment in our production advertising system, the code release requires strict internal review and approval processes to ensure compliance with company policies regarding intellectual property and business confidentiality. We have enhanced reproducibility by expanding the appendix with much more precise implementation details, including complete hyperparameter settings for all components, detailed architectural specifications, as well as training and inference protocols.
> We are committed to providing as much information as possible to enable the research community to understand and build upon our work, and we will make the code publicly available as soon as the internal review process is completed.
>
> ---
>
> Thank you again for your meticulous review. Your comments have directly led to plans for significant improvements in our manuscript. We hope the information provided has been helpful in addressing your concerns. Please don’t hesitate to reach out if you have any further questions or concerns!

---

> > ### Comment · Reviewer_yR4B · 2025-11-27
> >
> > Thanks for the response. I think the authors have addressed my concerns, for which I will increase the score.

---

> > > ### Author Response · Authors · 2025-11-27
> > > **Thank you very much !**
> > >
> > > Thank you very much for your positive recognition of our work and for the increased score. We sincerely appreciate your valuable time, effort, and insightful feedback, which helps us further improve our manuscript !
> > >
> > > Sincerely,
> > >
> > > The Authors

---

### Author Response · Authors · 2025-11-28
**General Reply**

We would like to express our sincere gratitude to all the Reviewers, ACs, SACs, and PCs for their valuable time, insightful feedback, and recognition of our work. The reviewers' recognition spans both methodological innovation (novel generative fusion, theoretical grounding, comprehensive experiments) and practical impact (computational efficiency, deployment feasibility, real-world gains).

We are particularly grateful that **Reviewer yR4B** promptly responded to our rebuttal on November 27, 2025 at 01:29 EST (converted from the original timestamp), **raising the score to 6 (before the bug-induced information leakage)**. We are deeply encouraged by the positive ratings and consistent suggestions for acceptance across all reviewers following our rebuttal.

We have thoroughly addressed all points raised during the discussion. Below, we summarize the key resolutions for the AC's convenience, highlighting how the concerns have been satisfied and solidified into a clear acceptance recommendation.

---

**For Reviewer yR4B & h94K (Theoretical Foundations & Empirical Validation):**
- **Feature Misalignment:** We clarified the inherent challenges of feature misalignment between long/short-term sequences, citing industrial constraints and related work. This explanation has been added to the revised manuscript.
- **Orthogonality in DCFG:** We provided both theoretical justification (via disentanglement loss) and new empirical evidence (low gradient cosine similarity <0.102) supporting the approximate orthogonality assumption in Theorem 1.
- **Theorem 2 & Mutual Information:** We elaborated on Theorem 2's proof and constants in Appendix E, and estimated session mutual information to validate the dependency conditions where our autoregressive injection excels.
- **Training Protocol:** We emphasized that single-epoch training represents the mainstream setup in CTR prediction research, supported by comprehensive multi-epoch analysis showing performance degradation in subsequent epochs across all methods, consistent with the well-documented one-epoch phenomenon in CTR literature.
- **Architecture Choices:** We explained our MLP-based design choice over attention variants as an optimal balance between performance and computational efficiency.

---

**For Reviewer n48p (Efficiency & Generalizability):**
- **Computational Efficiency:** We have thoroughly considered and evaluated these aspects in our work, with quantitative evidence provided throughout the paper. Our consistency loss enables efficient one-step inference, and we provide detailed complexity analysis showing minimal latency overhead (0.302% TP99 increase) and straightforward deployment, as illustrated in the deployment diagram in Appendix H.2 and the comprehensive efficiency analysis in Section 5.5.
- **Architecture Generalizability:** We have conducted extensive experiments to demonstrate iFusion's compatibility with various backbone architectures (DeepFM, DCN, AutoInt), consistently showing performance improvements over baseline fusion methods. These results are now included in the appendix of our revised paper.

---

**For Reviewer qLFV & h94K (Methodological Clarifications):**
- **Encoder Agnosticism:** We confirmed through ablation studies that iFusion's gains are consistent across different encoder architectures, establishing its modular nature.
- **Online Baseline:** We clarified that our online A/B test baseline was a highly-optimized, production-grade CTR prediction model serving as our primary industrial serving model, handling thousands of millions of users daily with rich sequential behavior signals. The statistically significant improvements against this powerful traditional discriminative model (CTR +2.44%, eCPM +2.61%) with minimal latency overhead provide compelling evidence for iFusion's practical value in real-world deployment.
- **MARN Terminology:** We clarified that "Mixture" refers to blending sequential guidance signals, not mixture-of-experts.
- **Robustness Validation:** We provided extensive analysis across special scenarios including promotional windows (behavioral shocks), cold-start sessions with masked data, and heterogeneous event mixtures, demonstrating iFusion's consistent performance and better robustness compared to discriminative baselines.

---

**Additional Improvements:**
- Enhanced reproducibility through expanded appendix with detailed implementation specifics
- Added comprehensive robustness analyses across multiple scenarios
- Provided complete multi-epoch training analysis

---

We deeply appreciate all reviewers, ACs, SACs, and PCs' constructive feedback, which has significantly strengthened our paper. The engaging discussions and thoughtful questions have helped us improve both the clarity and technical depth of our work. Thank you once again for your valuable time and consideration!

---

### Meta-Review · Area_Chair_kQgz · 2026-01-13

**Summary:**

The paper proposes iFusion, a diffusion-based generative framework for user interest fusion in CTR prediction. It introduces two key components: Disentangled Classifier-Free Guidance (DCFG) for separating core preferences from fluctuations, and Mixture AutoRegressive Denoising Network (MARN) for sequential interest modeling.

The reviewers unanimously recognized the novelty of reformulating interest fusion as a conditional generation process and the practical value demonstrated by extensive offline and online experiments.

Key concerns raised by reviewers during the review process included:

Theoretical Assumptions: Reviewers (yR4B, h94K) questioned the "approximate orthogonality" assumption in DCFG and the theoretical justification for MARN's autoregressive advantage.

Experimental Settings: Concerns were raised regarding the single-epoch training strategy (yR4B), the use of a simple DNN backbone instead of advanced architectures (n48p), and the lack of ablation on encoder choices (qLFV).

Efficiency and Deployment: Reviewers (n48p, h94K) were concerned about the inference latency typical of diffusion models and requested more details on the "consistency loss" trade-offs.

Motivation Clarification: Questions regarding the motivation for "feature misalignment" and the definition of "Mixture" in MARN.
Decision Rationale:

The authors provided a comprehensive rebuttal that addressed these concerns effectively. The consensus among reviewers is that the method is technically sound, novel, and practically viable. Given the empirical results and adequate theoretical clarifications, I recommend acceptance.

**Reviewer Concerns:**

Addressed Concerns:

Theoretical Validity (yR4B, h94K): The authors provided empirical evidence showing the average cosine similarity between core and fluctuation gradients is low, validating the orthogonality assumption in DCFG. They also clarified the proofs for Theorem 2 regarding the benefits of AR injection.

Experimental Rigor (yR4B, n48p):
Training Epochs: The authors demonstrated the "one-epoch phenomenon" prevalent in CTR tasks with new data showing performance degradation in subsequent epochs, justifying their setup.

Backbone Generalizability: New experiments showed iFusion consistently improves performance when integrated with advanced backbones like DeepFM, DCN, and AutoInt.

Efficiency (n48p, h94K): The authors quantified the cost, showing only a ~0.3% increase in offline inference time and a 0.302% increase in online TP99 latency. They also demonstrated that one-step inference (via consistency loss) yields optimal performance-efficiency trade-offs compared to multi-step inference.

Component Clarifications (qLFV, h94K): The authors clarified that the method is encoder-agnostic (via ablations) and that the online baseline is a highly optimized industrial model. They also clarified "Mixture" refers to blending sequential signals, not a Mixture-of-Experts architecture.

Outstanding Concerns:

Code Reproducibility: While the authors expanded the appendix with implementation details and promised a code release, the code is not yet public due to internal review processes. However, this is a common constraint for industrial papers and the added details significantly mitigate this.

**Reviewer Scores:**

Reviewer yR4B: This reviewer actively participated and explicitly raised their score after the authors clarified the orthogonality issues and experimental settings.

Reviewer n48p: Maintained a positive score. The addition of generalizability experiments and efficiency analysis likely solidified their positive stance.

Reviewer qLFV: Maintained a positive score. Their concerns about encoders and baselines were fully resolved.

Reviewer h94K: Maintained a positive score. The clarifications on DCFG intuition and zero-data scenarios strengthened their assessment.

Overall, the scores have converged to a consensus for acceptance.

---

### Decision · Program_Chairs · 2026-01-26

Accept (Poster)